# Unique ocean circulation pathways reshape the Indian Ocean oxygen minimum zone with warming

Sam Ditkovsky[1], Laure Resplandy[2], and Julius Busecke[3]

[1]Program in Atmospheric and Oceanic Sciences, Princeton University, Princeton, NJ, USA
[2]Department of Geosciences and High Meadows Environmental Institute, Princeton University, Princeton, NJ, USA
[3]Lamont-Doherty Earth Observatory, Columbia University, New York, NY, USA

**Correspondence:** Sam Ditkovsky (samjd@princeton.edu)

**Abstract.** The global ocean is losing oxygen with warming. Observations and Earth system model projections suggest, however, that this global ocean deoxygenation does not equate to a simple and systematic expansion of tropical oxygen minimum zones (OMZs). Previous studies have focused on the Pacific Ocean; they showed that the outer OMZ deoxygenates and expands as oxygen supply by advective transport weakens, the OMZ core oxygenates and contracts due to a shift in the composition of the source waters supplied by slow mixing, and in between these two regimes, oxygen is redistributed with little effect on OMZ volume. Here, we examine the OMZ response to warming in the Indian Ocean using an ensemble of Earth system model high-emissions scenario experiments from the Coupled Model Intercomparison Project phase 6. We find a similar expansion-redistribution-contraction response, but show that the unique ocean circulation pathways of the Indian Ocean leads to far more prominent OMZ contraction and redistribution regimes than in the Pacific Ocean. As a result, only the outermost volumes (oxygen $> 180\,\mu\mathrm{mol/kg}$) expand. The Indian Ocean experiences a broad oxygenation in the southwest driven by a reduction in waters supplied by the Indonesian Throughflow in favor of high-oxygen waters supplied from the South Indian Gyre. Models also project a strong localized deoxygenation in the northern Arabian Sea due to the rapid warming and shoaling of marginal sea outflows (Red Sea and Persian Gulf), and increases in local stratification with warming. We extend the existing conceptual framework used to explain the Pacific OMZ response to interpret the response in the Indian Ocean.

## 1 Introduction

Oxygen minimum zones (OMZs) are naturally occurring low-oxygen regions located in subsurface tropical oceans (typically 100-1500 m). OMZs develop in the "shadow zones" of the ocean thermocline where oxygen supply by ocean circulation is weak (Luyten et al., 1983; Pedlosky, 1983), and are generally located below highly productive surface systems that boost respiration and biological oxygen demand at the subsurface (Paulmier and Ruiz-Pino, 2009). The global ocean has lost oxygen in response to global warming (Keeling et al., 2010; Helm et al., 2011; Schmidtko et al., 2017; Bindoff et al., 2019), and this trend is expected to continue and accelerate over the twenty-first century if anthropogenic emissions are not significantly drawn

down (Bopp et al., 2013; Kwiatkowski et al., 2020). Global deoxygenation has been attributed to weakening ocean ventilation (i.e., weakening oxygen supply by ocean circulation and mixing) and decreasing oxygen solubility in seawater with warming (e.g., Oschlies et al., 2018). A concern is that OMZs are expanding in response to global deoxygenation, potentially disrupting the physiology and survival of marine organisms and compressing the habitats of marine species requiring oxygen for their survival (Vaquer-Sunyer and Duarte, 2008; Miller et al., 2002; Stramma et al., 2008, 2012; Deutsch et al., 2020; Levin, 2018). The fate of OMZs under warming, in particular the two most intense regions found in the tropical Pacific Ocean and tropical Indian Ocean, has however been highly debated, with apparently inconsistent changes found across hydrographic observations, paleo-oceanographic proxies and Earth system model (ESM) projections. In-situ hydrographic observations collected since the 1950s suggest that the tropical Indo-Pacific Oceans and the marginal seas that supply oxygen to the tropical Indian Ocean (including the Persian Gulf and Red Sea) have lost oxygen, supporting the view that tropical OMZs are expanding (Stramma et al., 2008; Helm et al., 2011; Ito et al., 2017; Banse et al., 2014; Piontkovski and Al-Oufi, 2015; Queste et al., 2018; Naqvi, 2021). Yet, paleo-oceanographic studies suggest that the OMZ in the eastern tropical Pacific Ocean has contracted, rather than expanded, under past warming conditions (Deutsch et al., 2014; Auderset et al., 2022). Looking into the future, studies using ESM ensembles have projected a robust deoxygenation at mid- and high-latitudes with warming, consistent with the weakening of ventilation found at the global scale, but have failed, until recently, to reach a consensus on the expected changes in oxygen and OMZ volumes in tropical oceans (Cocco et al., 2013; Bopp et al., 2013; Cabré et al., 2015; Bopp et al., 2017; Resplandy, 2018; Kwiatkowski et al., 2020).

Busecke et al. (2022) recently showed that the inconsistencies found in the fate of the OMZ in the Pacific Ocean could be reconciled using an ensemble of ESMs from the Coupled Model Intercomparison Project Phase 6 (CMIP6; Eyring et al., 2016). They found that the OMZ response to global warming was in fact consistent across the ESMs when examined in an oxygen-space framework and fell into three regimes: an expansion of the OMZ outer layers (large OMZ volume delimited by oxygen thresholds of typically $\sim100\,\mu\mathrm{mol/kg}$ or higher), a contraction of the eastern Pacific "OMZ core" waters (OMZ volume delimited by oxygen thresholds of $\sim20\,\mu\mathrm{mol/kg}$ or lower), and a "transition regime" between contraction and expansion that experiences weak and uncertain changes associated with a spatial redistribution of the OMZ volume. This three-regime framework reconciles hydrographic work that show an OMZ expansion in the central Pacific Ocean where the OMZ outer layers are located (Stramma et al., 2008), and paleo-oceanographic studies that found evidence for a contraction in the eastern Pacific (note that these studies used nitrogen isotopes which are a proxy for "OMZ core" denitrifying waters; Deutsch et al., 2014; Auderset et al., 2022). The framework also explains the discrepancies found in previous modeling studies that often considered OMZ volume definitions that fall in the transition regime where changes are smaller and uncertain (e.g., Bopp et al., 2013; Cabré et al., 2015). In the Pacific Ocean, the apparent discrepancy between outer OMZ expansion and core contraction can be interpreted using the conceptual framework proposed by Gnanadesikan et al. (2007), which distinguishes between two models of ocean ventilation: (1) a "single-pipe" model where ventilation is controlled by advection from one source water mass, and (2) a "mixing-network" model in which ventilation rates are sustained by slow mixing of multiple source waters as in ocean shadow zones (Lévy et al., 2022; Gnanadesikan et al., 2013; Brandt et al., 2015). As ocean circulation pathways weaken, regions ventilated by a single-pipe experience reduced supply of oxygen (transport rates of oxygenated surface water to the

thermocline slows), while regions ventilated by a mixing-network can either experience a reduction or increase in ventilation and oxygen supply (by changing the connectivity and the contributions of each source water to the mixing-network). The single-pipe model explains the deoxygenation of the outer OMZ layers, which has been attributed to the weakening of the northern and southern subtropical cells that ventilate the outer layers of the OMZ (Gnanadesikan et al., 2012; Duteil et al., 2014; Busecke et al., 2022; Duteil et al., 2021; Llanillo et al., 2018; Margolskee et al., 2019). In contrast, changes in mixing-network connectivity explain the oxygenation of the Pacific OMZ core, which was attributed to reduced contributions from aged, oxygen-poor deep and intermediate waters, shifting the mixing ratio towards younger, oxygen-rich upper ocean waters (Bryan et al., 2006; Gnanadesikan et al., 2007, 2012; Takano et al., 2018; Busecke et al., 2022).

The fate of the Indian Ocean OMZ has been far less studied than its Pacific Ocean counterpart. Yet, the Indian Ocean shows some of the fastest ocean warming trends in the world (Roxy et al., 2014; Sharma et al., 2023), and the expansion of its OMZ could have detrimental effects for coastal populations that depend heavily on marine resources for food security and economic stability in the region (Bouchard and Crumplin, 2010; Clifton et al., 2012; Gattuso et al., 2015; Llewellyn et al., 2016; Roy, 2019). The ventilation pathways and OMZ geometry in the Indian Ocean are fundamentally different from the two subtropical cells (one in each hemisphere) and eastern boundary OMZ found in the Pacific Ocean. It is thus unclear whether the "single-pipe and mixing-network" framework which characterizes the Pacific Ocean is sufficient to describe the behavior of the Indian Ocean. The Indian Ocean is bounded by continent to the north and ventilation is almost exclusively sustained by one subtropical cell originating in the South Indian Gyre (Harper, 2000; Schott et al., 2002; Phillips et al., 2021). As a result, oxygen levels are higher in the southern Indian Ocean while an OMZ extends over most of the Arabian Sea and the Bay of Bengal in the north (see Fig. 1 for observed climatological oxygen field and major ventilation pathways). Other peculiarities of the Indian Ocean ventilation are the Indonesian Throughflow which brings waters from the tropical Pacific Ocean into the Southern Indian Ocean (Sprintall et al., 2009), and the saline marginal sea outflow waters from the Red Sea and Persian Gulf that ventilate the Arabian Sea (Rhein et al., 1997; Beal et al., 2000; Menezes, 2021; Sheehan et al., 2020). Ventilation by all these advective pathways (Southern Gyre, Indonesian Throughflow and marginal seas) is projected to weaken in response to climate change (Sen Gupta et al., 2016; Feng et al., 2017; Stellema et al., 2019; Lachkar et al., 2019; Kobayashi et al., 2012), but the extent to which these changes in ventilation will affect basin-scale oxygen content and the OMZ in the Indian Ocean is however still poorly constrained. Here, we examine changes in oxygen content and OMZ volume in the Indian Ocean in response to climate change using an ensemble of CMIP6-generation ESMs with a focus on thermocline depths (upper 1000 m). We show that the fate of the Indian Ocean OMZ is consistent with the three-regime framework identified in the Pacific Ocean (Busecke et al., 2022) but that broad oxygenation results in a much more prominent contraction regime in the Indian Ocean than in the Pacific Ocean. Interpreting these changes in oxygen and OMZ volume in the Indian Ocean calls for an extension of the single-pipe and mixing-network conceptual framework to include the unique contributions from the Indonesian Throughflow and marginal seas, which we show can be interpreted as "two-pipe" and "moving-pipe" systems.

## 2 Methods

### 2.1 Datasets

We use an ensemble of eight ESMs from the CMIP6 archive (Eyring et al., 2016; O'Neill et al., 2016). Out of the 14 CMIP6 ESMs that provided monthly dissolved oxygen data for the pre-industrial control, historical, and SSP5-8.5 experiments (Busecke et al., 2022), we exclude six models that simulate virtually no suboxic ($\leq$10 $\mu$mol/kg) volume in the Arabian Sea (ACCESS-ESM1-5, CanESM5, CanESM5-CanOE, CNRM-ESM2-1, IPSL-CM6A-LR; Table S1). We keep the eight remaining ESMs (GFDL-CM4, GFDL-ESM4, MIROC-ES2L, MPI-ESM1-2-HR, MPI-ESM1-2-LR, NorESM2-LM, NorESM2-MM, UKESM1-0-LL). All six ESMs excluded from the multi-model mean exhibit above average salinity biases in the Arabian Sea, likely from outflows (Fig. S1), and four of six models exhibit Red Sea outflow rates over twice the observed rate (Fig. S2). Thus, the representation of marginal sea outflows may be improved significantly in our ensemble by excluding these ESMs. For this analysis, we use oxygen, salinity, and potential temperature for all 8 models. When available, we also use output for ideal age (available for six ESMs), export of organic carbon at 100 m (available for seven ESMs) and mass transport (available for 5 ESMs). Stratification is calculated from potential temperature and salinity fields using the `GSW-python` package (Firing et al., 2021). To limit computational costs of this study, and because we expect inter-model variability in dissolved oxygen to dominate over internal variability, only one member is used for each model. See Table 1 for member labels and data availability of each model. All model outputs were regridded (via bilinear interpolation) to a uniform $1° \times 1°$ grid using the xESMF python package (Zhuang et al., 2021), but transport calculations were performed on each model's native grid (Sect. 2.2.4). The pre-industrial simulations were used to remove the linear control drifts from all scalar fields (oxygen, salinity, temperature, ideal age, export) in the historical and SSP5-8.5 simulations using the xMIP python package (Busecke and Spring, 2020).

We also use the observational climatology of dissolved oxygen concentrations from the World Ocean Atlas 2018 (WOA18; Garcia et al., 2019) to evaluate the representation of the Indian Ocean OMZ and dissolved oxygen field in the ensemble of ESMs. We use an average over the period of 1950-2015 in the historical simulations to compare to the observed climatology.

### 2.2 Analysis

To characterize the response of ocean variables to climate change, we used linear trends over the 2015-2100 period in the SSP5-8.5 simulations (normalized to change per century), except when calculating water mass fractions (Sect. 2.2.3), for which we compare historical (1950-2015 average) and end-of-century (historical plus integrated linear trend over 85-years) states. To represent the ESM ensemble, we take the mean over models (multi-model mean) and use one standard deviation of the model spread on either side of the mean as a measure of uncertainty. When presenting multi-model mean trend fields, we stipple where less than 75 % of available models agree on the sign of change to indicate regions of uncertainty. While we perform our analysis over the full water column, we focus on thermocline depths (upper 1000 m) so that the results may characterize impacts on mesopelagic ecosystems. When examining oxygen and ideal age trends, we exclude surface waters by examining changes between 100 and 1000 m.

**Table 1.** ESM data used in this study. Variables used: Dissolved oxygen concentration (o2), salinity (so), potential temperature (thetao), ideal age (agessc), export of organic carbon at 100 m (epc100) and mass transport (umo/vmo) where available. All data used in this study is publicly available via ESGF, except ideal age fields from GFDL-CM4 and GFDL-ESM4 (see Busecke et al., 2022)

| ESM | Variables | Member ID |
| --- | --- | --- |
| GFDL-CM4 (Guo et al., 2018a, b) | o2, thetao, so, agessc | r1i1p1f1 |
| GFDL-ESM4 (Krasting et al., 2018; John et al., 2018) | o2, thetao, so, agessc, epc100 | r1i1p1f1 |
| MIROC-ES2L (Hajima et al., 2019; Tachiiri et al., 2019) | o2, thetao, so, agessc, epc100 | r1i1p1f2 |
| MPI-ESM1-2-HR (Jungclaus et al., 2019; Schupfner et al., 2019) | o2, thetao, so, epc100, umo/vmo | r1i1p1f1 |
| MPI-ESM1-2-LR (Wieners et al., 2019b, a) | o2, thetao, so, agessc, epc100, umo/vmo | r1i1p1f1 |
| NorESM2-LM (Seland et al., 2019a, b) | o2, thetao, so, agessc, epc100, umo/vmo | r1i1p1f1 |
| NorESM2-MM (Bentsen et al., 2019a, b) | o2, thetao, so, agessc, epc100, umo/vmo | r1i1p1f1 |
| UKESM1-0-LL (Tang et al., 2019; Good et al., 2019) | o2, thetao, so, epc100, umo/vmo | r1i1p1f2 |

### 2.2.1 Tracking ocean volume by oxygen threshold

OMZ volume is generally defined as the volume of water below a chosen oxygen concentration threshold. We extend this idea to a wide range of oceanic oxygen concentration values. Here, we define the full column OMZ volume as a function of oxygen thresholds, $O_{2,T}$ (following Busecke et al., 2022):

$$\mathcal{V}_{O_2}(O_{2,T}) = \iiint_{O_2 \leq O_{2,T}} dV \tag{1}$$

where we integrate over the Indian Ocean from $30°\,S$ - $25°\,N$ and from the African continent to the west to Indonesia and Australia, extending to about $125°\,E$ in the main strait of the Indonesian Throughflow (Timor Sea), to the east. The Red Sea and Persian Gulf are not included in the integration of $\mathcal{V}_{O_2}$. We use the Equator to delimit the Arabian Sea and Bay of Bengal sub-basins to the south. However, we do not focus on the full OMZ volume, but rather on the thermocline OMZ volume:

$$\mathcal{V}_{O_2^{1000}}(O_{2,T}) = \int_0^{1000m} \iint_{O_2 \leq O_{2,T}} dV \tag{2}$$

Note that we integrate over the upper 1000 m here, rather than 100 to 1000 m, because OMZ volume will naturally exclude surface waters. In both cases, we use oxygen thresholds $O_{2,T}$ that vary between 5 and $225\,\mu mol/kg$ and highlight three benchmark thresholds: (1) $O_{2,T} = 20\,\mu mol/kg$ (OMZ20) as the core of the OMZ, (2) $O_{2,T} = 60\,\mu mol/kg$ (OMZ60) as a commonly cited threshold for hypoxia at which marine ecosystems tend to experience significant loss of biodiversity (Vaquer-Sunyer and Duarte, 2008), and (3) $O_{2,T} = 150\,\mu mol/kg$ (OMZ150) as a common habitat boundary for large commercial fish species such as tuna (Brill, 1996; Prince and Goodyear, 2006; Bertrand et al., 2011; Stramma et al., 2012; Dueri, 2017). We note that oxygen thresholds above $150\,\mu mol/kg$ are not generally used to delimit low oxygen environments and OMZs, but we present them here to provide a holistic view of forced changes in oxygen distribution.

### 2.2.2 Thermal and non-thermal dissolved oxygen trends

We separate the influences of thermal and non-thermal processes on oxygen trends. The thermal component, or oxygen saturation $O_{2SAT}$, is calculated using the `GSW-python` package from potential temperature and salinity fields (Firing et al., 2021). The non-thermal component, related to ocean ventilation and biological sources and sinks of oxygen, is computed as the residual between $O_2$ and $O_{2SAT}$ and is referred to as Apparent Oxygen Utilization (AOU):

$$O_2 = O_{2SAT} - AOU \tag{3}$$

We compute $O_{2SAT}$ and AOU for each year in the historical and SSP5-8.5 simulations, then take historical means and projected linear trends. Trends in $-$AOU represent the contribution of non-thermal processes to overall changes in dissolved oxygen but they encompass both changes in physical (ventilation changes tied to slower circulation or changes in mixing) and biological (respiration rates) effects. We use the ideal age of seawater (i.e., the time since exposure to the surface) to qualitatively infer the contribution of physical ventilation changes.

### 2.2.3 Identifying changes in ventilation pathway contributions

In sections 3.5 and 3.6, we solve a mixture model in the South Equatorial Current and in the Arabian Sea to evaluate shifts in water mass fractions under SSP5-8.5 forcing. The mixture model is based on extended Optimum Multiparameter Analysis (OMP) (Tomczak Jr, 1981; Tomczak and Large, 1989; Karstensen and Tomczak, 1998). This technique solves for the water mass fractions at a given hydrographic section using the hydrographic properties of specified source water mass types, each representing an advective ventilation pathway. See Appendix A and references therein for details on the method, and how the mixture model implemented here differs from classical OMP. We use a python implementation (`pyOMPA`; Shrikumar et al., 2022), based on the original MATLAB code from Karstensen and Tomczak (1998). We perform this analysis along potential density surfaces referenced to zero pressure ($\sigma_0$), calculated from potential temperature and salinity fields using the `GSW-python` package (Firing et al., 2021). For each region, we use a nominal value for potential density that captures features of significant dissolved oxygen change; however, we tune this value (within 0.5) for individual ESMs to better capture features across models.

The number of source water mass types is limited to be less than or equal to the number of hydrographic properties available. Here, we use potential temperature, salinity, dissolved oxygen and AOU. To account for remineralization along the pathways (between source and case study regions), we allow for the conversion of dissolved oxygen to AOU (see Appendix A for details). In the case of the South Equatorial Current region, the source water types are Indonesian Throughflow Water, Southern Pathway Waters and Arabian Sea Water, centered at ($15°$ S, $120°$ E), ($30°$ S, $100°$ E) and ($15°$ N, $65°$ E) respectively at a nominal potential density of $\sigma_0 = 26.4$. In the case of the Arabian Sea, the source waters are Persian Gulf Water, Red Sea Outflow Water and Arabian Sea Water centered at ($24.5°$ N, $58.5°$ E), ($12°$ N, $47°$ E) and ($15°$ N, $65°$ E) respectively at a nominal potential density of $\sigma_0 = 25.7$. We exclude MIROC-ES2L from the mixture model analysis in the Arabian Sea because the model does not resolve the Persian Gulf and Red Sea. To account for the sensitivity of results to small changes in the sampling of source water

types and potential density layer, we average 50 realizations of the experiment applying random perturbations to source water locations and density layers from their central values. Source water locations are perturbed by up to $5°$ in latitude and longitude for the South Equatorial Current region and $2°$ for the Arabian Sea. For both locations, the value of the potential density layer is perturbed by up to $0.1 \ \mathrm{kg/m^3}$.

For each ESM in the ensemble, we solve for the water mass fractions, $f$, for a historical state and an end-of-century state. For the historical state, we first compute potential density for the historical mean (1950-2015) for each ESM, and use the `xgcm` (Abernathey et al., 2022) package to transform the vertical coordinates of each model to potential density space. We note that the results of the mixture model are not sensitive to the order of operations (temporal averaging and vertical transformation). To compute an end-of-century state, we add 85-year (2015-2100) forced trends from the SSP5-8.5 simulations to historical mean fields (potential temperature, salinity, oxygen, AOU) to represent a climatological year 2100, then repeat the coordinate transformation described above.

From the mixture model results, we quantify the change in oxygen supply due to shifts in water mass composition and properties for key regions:

$$\Delta O_{2,\mathrm{supply}} = \sum_i f_{i,1} O_{i,1} - \sum_i f_{i,0} O_{i,0} \tag{4}$$

where $f_i$ and $O_i$ are the fraction and oxygen concentration of source water type $i$, for future (1) and historical (0) states. We compare $\Delta O_{2,\mathrm{supply}}$ to the total simulated changes in oxygen to evaluate how well changes in the system can be described by only isopycnal transport and mixing of source water mass types. While for conservative tracers (i.e. potential temperature and salinity) the change in supply and total change should be equal if the system is well represented by the mixture model analysis, an allowance for the remineralization of oxygen can lead to an imbalance between the two quantities (see Appendix A).

A concern when solving for forced trends on a potential density layer is that the results may be influenced by the displacement of this isopycnal in depth. Indeed, the potential density layers chosen here are displaced deeper in the water column by about 100 m between the historical and end-of-century states. To evaluate the impact of this deepening, we perform an alternate set of mixture model experiments where we use a lower density value at the end-of-century state than for evaluating the historical state. The layer defined by this lower density value samples the same depth as was sampled for the historical state. The results of this alternative analysis, presented in Supplementary Information, are qualitatively similar to the analysis presented in the main text, and do not change the main results.

### 2.2.4 Quantifying changes in transport by ventilation pathways

To provide context for trends in dissolved oxygen and ideal age, we quantify changes in individual thermocline ventilation pathways using mass transport fields on each model's native grid. The Indonesian Throughflow Water transport is calculated as the westward flow through the main strait of the Indonesian Throughflow at $114°E$ integrated between 100 and 1000 m. The Southern Pathway Waters transport is calculated as the northward flow across $30°S$ integrated between 100 and 1000 m. We note that this representation of Southern Pathway Waters largely excludes the contribution of Subtropical Underwaters,

which subduct north of 30°S. The upwelling of Deep Waters into the thermocline is calculated as the budget residual of transport below 1000 m across 30°S and through the Indonesian Throughflow at 114°E. Transports by marginal sea outflows from the Red Sea and Persian Gulf are defined as the outflowing components through their respective channels. We report transport trends in Sv/century, using a constant reference density of $1025 \, \mathrm{kg/m^3}$ to convert between mass and volume transport. Transport fields are available for five of the eight ESMs in the ensemble (MPI-ESM1-2-HR, MPI-ESM1-2-LR, NorESM2-LM, NorESM2-MM, UKESM1-0-LL).

## 3    Results

### 3.1    Historical oxygen and ventilation pathways in observations and Earth system models

The observed oxygen distribution in the Indian Ocean thermocline (100 to 1000 m) is influenced by five main ventilation pathways (Fig. 1a-c). First, the Southern Pathway Waters which enter via the South Indian Gyre include (from least to most dense) Subtropical Underwater (STUW; about 100 to 250 m depth), Indian Central Water and Subantarctic Mode Water (CW+MW; about 250 to 750 m depth) and Antarctic Intermediate Water (IW; about 750 to 1000 m depth) (Sprintall and Tomczak, 1993; Karstensen and Tomczak, 1997; McCarthy and Talley, 1999; Karstensen and Quadfasel, 2002; Fine, 1993; Talley, 2011). This spectrum of waters enters the basin with a range of oxygen between about 160 to $250 \, \mu\mathrm{mol/kg}$, where Indian Central Water and Subantarctic Mode Water account for the highest oxygen levels. Southern Pathway Waters are capped by the salinity maximum of Subtropical Underwater and the salinity minimum of Antarctic Intermediate Water (dashed black contours in Fig. 1b,c). Second, the Indonesian Throughflow (ITF) brings waters from the tropical Pacific Ocean (with oxygen concentrations of about 80-170 $\mu\mathrm{mol/kg}$ between 100 to 1000 m depth) into the Southern Indian Ocean and which mix with Southern Pathway Waters in the South Equatorial Current (SEC). This mixture of waters then crosses the equator with the western boundary current to ventilate the northern basin (pathway schematic in Fig. 1a) via eddy mixing, monsoonal currents and the weak North Equatorial Current which flows east in the summer along about 5° N (Schott et al., 2009; Phillips et al., 2021; Resplandy et al., 2012; Lachkar et al., 2016). In the northern Indian Ocean, two additional pathways are associated with saline marginal sea outflows: Red Sea Outflow Water (RS, with concentrations around $70 \, \mu\mathrm{mol/kg}$) and Persian Gulf Water (PG, near oxygen saturation >$200 \, \mu\mathrm{mol/kg}$) deliver oxygen directly to the OMZ core in the Arabian Sea as they are quickly diluted by mixing (Fig. 1a). Finally, the fifth pathway corresponds to Deep Waters (DW) that enter from the Southern Ocean and Indonesian Throughflow (here taken as waters below 1000 m depth) and slowly upwell in the basin (McCarthy et al., 1997), ventilating the thermocline from below with relatively well oxygenated waters (100 - $200 \, \mu\mathrm{mol/kg}$).

The multi-model mean (eight ESMs, see methods) captures the main features associated with the five pathways ventilating the Indian Ocean (Fig. 1d-f). The multi-model mean simulates the full spectrum of Southern Pathway Waters, indicated by the salinity maximum and minimum associated with Subtropical Underwater and Antarctic Intermediate Water (dashed black contours in Fig. 1e,f), and overall captures the temperature and salinity properties of water masses across the basin (Fig. S5). We note that three of the eight ESMs simulate the Southern Pathway Waters particularly well (GFDL-CM4, GFDL-ESM4, and UKESM1-0-LL), while three others are missing the salinity minimum associated with Antarctic Intermediate

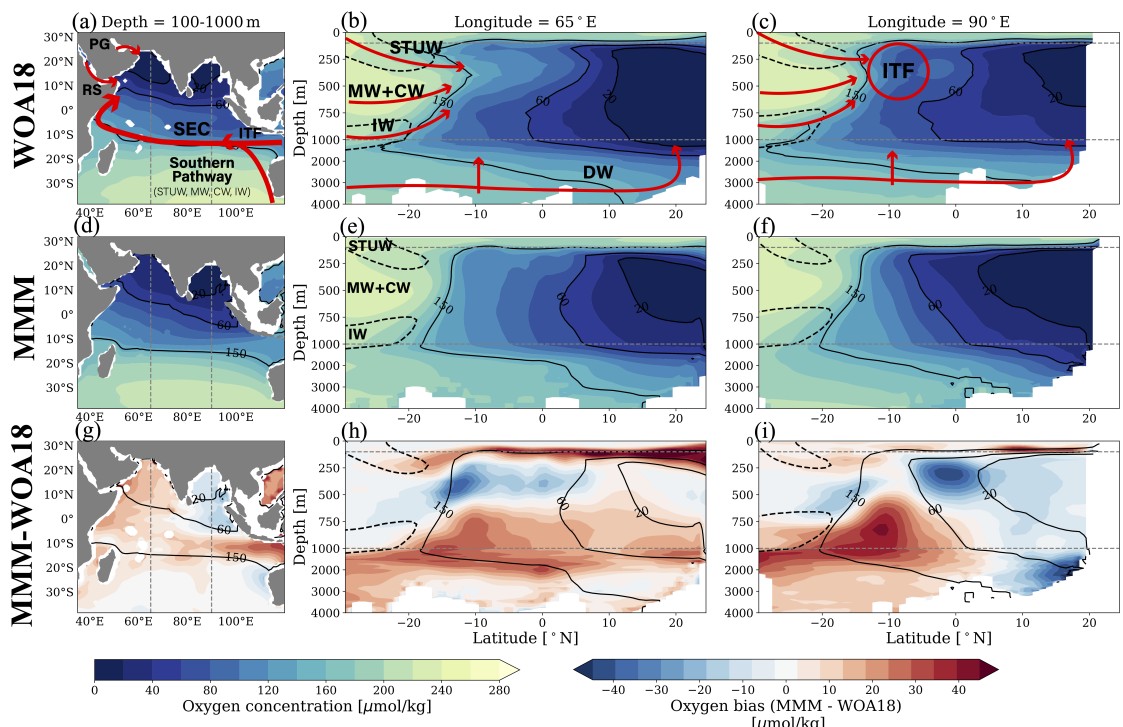

**Figure 1.** Dissolved oxygen annual mean climatology from World Ocean Atlas 2018 (WOA18) **(a)** averaged between 100 and 1000 m, at **(b)** 65°E and **(c)** 90°E. Multi-model mean (MMM) dissolved oxygen in the Indian Ocean for historical period (1950-2015) **(d)** between 100 and 1000 m, **(e)** at 65°E, and **(f)** at 90°E. Difference between MMM (1950-2015) and WOA18 dissolved oxygen **(g)** between 100 and 1000 m, **(h)** at 65°E, and **(i)** at 90°E. Solid black contours represent 20, 60 and 150 μmol/kg oxygen in **(a-c)** WOA18 and **(d-i)** MMM. **(a,d,g)** Dashed gray lines indicate 65°E and 90°E. **(b,c,e,f,h,i)** Dashed gray lines indicate depth of 100 and 1000 m, dashed black contours show salinity contours highlighting Subtropical Underwater and Intermediate Water. Water masses and ventilation pathways are illustrated schematically. Abreviations: South Equatorial Current (SEC), Central Water (CW), Deep Water (DW), Subtropical Underwater (STUW), Mode Water (MW), Intermediate Water (IW), Indonesian Throughflow (ITF).

Water (MIROC-ES2L, MPI-ESM1-2-HR and MPI-ESM1-2-LR, Fig. S4). The oxygen supply from the Southern Pathway and the Indonesian Throughflow is also relatively well simulated by the multi-model mean (Fig. 1a-c), and the observed oxygen maximum associated with Indian Central Water and Subantarctic Mode Water is present in all the ESMs in the ensemble
(though it is weaker than observed in MPI-ESM1-2-HR and MPI-ESM1-2-LR, see Fig. S3, S4). Overall, these simulated thermocline ventilation pathways maintain a realistic meridional oxygen gradient with well oxygenated waters in the southern basin and OMZ cores in the Arabian Sea and Bay of Bengal (contours of 20, 60 and 150 μmol/kg water extent shown in solid black contours in Fig. 1 and Fig. S4). There are, however, significant biases in the oxygen field simulated in northern basin where the multi-model mean simulates higher than observed oxygen levels in the Arabian Sea, and lower than observed in
the Bay of Bengal (Fig. 1g-i and Fig. S3). The high oxygen bias in the Arabian Sea is likely due to excessive ventilation from

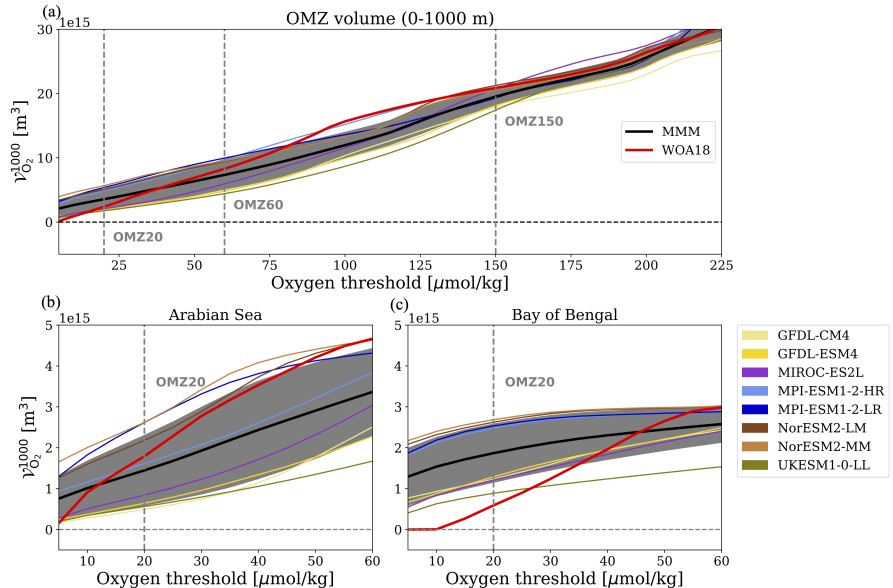

**Figure 2.** OMZ volume taken between 0-1000 m in the **(a)** Indian Ocean and sub-basins **(b)** Arabian Sea and **(c)** Bay of Bengal for the multi-model mean averaged over 1950-2015 (MMM; black) and observed climatology (WOA18; red). Shading represents one standard deviation of the model spread from the mean. Individual ESMs shown in colored curves.

marginal seas and southern source waters, though may also be influenced by deficiencies in parameterized oxygen consumption rates and eddy-mixing rates (Schmidt et al., 2021), while the low oxygen bias in the Bay of Bengal is possibly tied to the overestimation of oxygen consumption rates and remineralization depths (Al Azhar et al., 2017). These are well documented biases among ESMs (Oschlies et al., 2008; Bopp et al., 2013; Rixen et al., 2020; Schmidt et al., 2021). However, we have mitigated oxygen biases in the Arabian Sea in our ensemble by removing models that lack suboxia there (Sect. 2.1; Fig. S6).

We compare thermocline OMZ volume ($\mathcal{V}_{\mathrm{O}_2^{1000}}$, volume in upper 1000 m) in observations and the multi-model mean for oxygen thresholds between 5 and 225 $\mu$mol/kg (Fig. 2a, $\mathcal{V}_{\mathrm{O}_2^{1000}}$ for individual models are in Table S1.). In the observed climatology, $\mathcal{V}_{\mathrm{O}_2^{1000}}$ increases about linearly with oxygen threshold, with an hypoxic volume OMZ60 ($<60\,\mu$mol/kg) of about $8.3\times10^{15}\mathrm{m}^3$ and a low oxygenated water volume OMZ150 ($<150\,\mu$mol/kg) of about $21\times10^{15}\mathrm{m}^3$. The multi-model mean simulates this linear relationship. In particular, it agrees with observations within 12 % for the OMZ60 volume and within 7 % for the OMZ150 volume, and at both benchmarks the observations fall within the ensemble spread with no clearly outlying ESM. The volume of the core OMZ20 is, however, overestimated by about 50 % in the multi-model mean. The good agreement at hypoxic and higher thresholds arises from the realistic simulation of the southern Indian Ocean ventilation pathways that control these volumes. In contrast, the disagreement in the volume of the OMZ core primarily comes from the systematic overestimation of OMZ20 in the Bay of Bengal, a bias only partially offset by the underestimation of OMZ20 in the Arabian Sea (Fig. 2b,c).

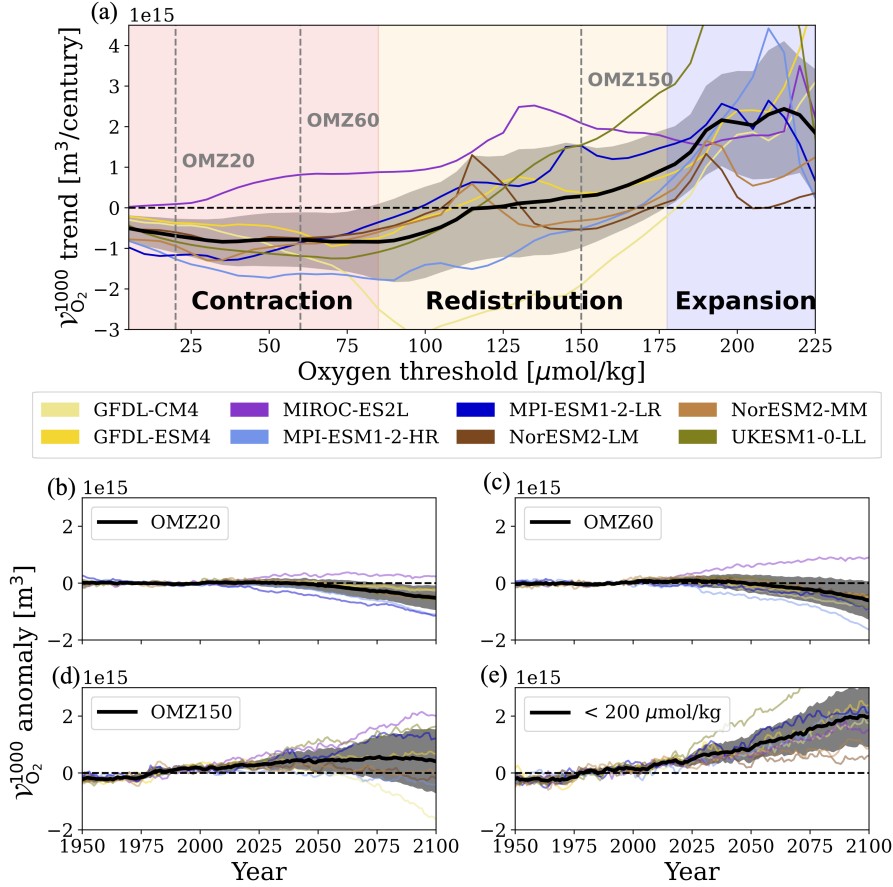

**Figure 3.** Multi-model mean thermocline OMZ volume changes (between 0 - 1000 m) under SSP5-8.5 scenario forcing (2015-2100). **(a)** $\mathcal{V}_{O_2^{1000}}$ trends as a function of oxygen threshold for the Indian Ocean. The 20, 60 and 150 $\mu$mol/kg thresholds bounding OMZ20, OMZ60 and OMZ150 are indicated with gray dashed lines. Time series of $\mathcal{V}_{O_2^{1000}}$ anomaly from 1950-2100 (anomaly referenced to 1950-2015 mean) for **(b)** OMZ20, **(c)** OMZ60 and **(d)** OMZ150, and **(e)** waters with $< 200\,\mu$mol/kg oxygen. Shading represents one standard deviation of model spread from the mean.

### 3.2  Projected OMZ volume exhibits diverging trends at low and high oxygen thresholds

Projections of thermocline OMZ volume under SSP5-8.5 forcing fall into three regimes: a robust expansion of volumes set by high oxygen thresholds (above 180 $\mu$mol/kg), a robust contraction of volumes set by low oxygen thresholds (below

85 $\mu$mol/kg), and a transition regime with weak and uncertain trends characterized by redistribution of volume (between 85 and 180 $\mu$mol/kg; Fig. 3a). We define robust OMZ expansion and contraction as where the multi-model mean trends exceed one standard deviation of the model spread. The volumes of hypoxic waters (OMZ60) and core waters (OMZ20) contract at similar rates in the multi-model mean ($0.7\pm0.4\times10^{15}\,\mathrm{m}^3$/century and $0.8\pm0.7\times10^{15}\,\mathrm{m}^3$/century, respectively). The model

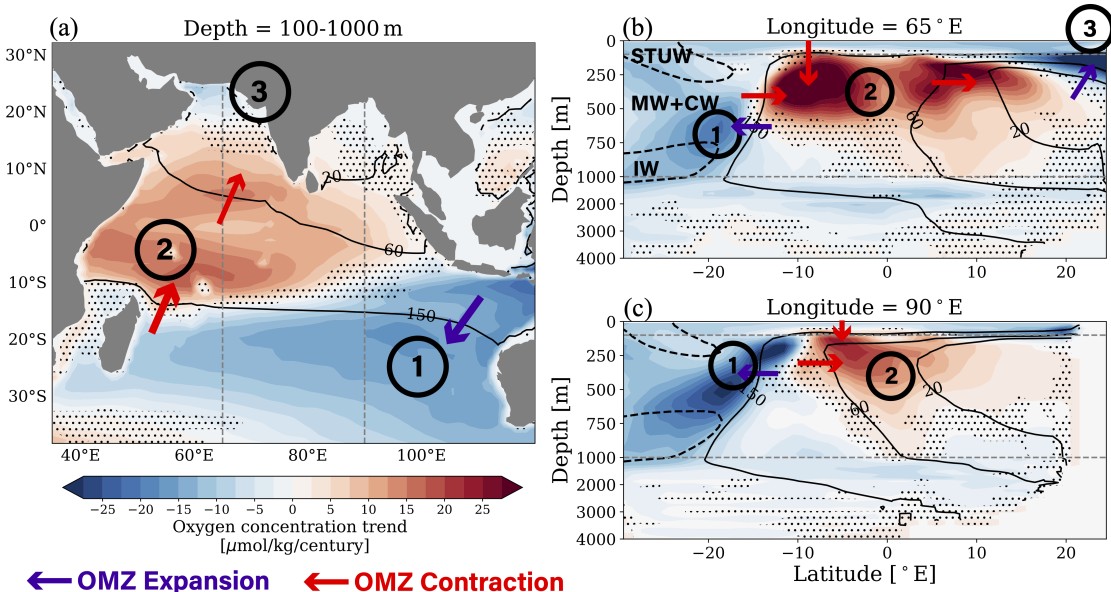

**Figure 4.** Multi-model mean (MMM) dissolved oxygen trends under SSP5-8.5 scenario forcing (2015-2100). Dissolved oxygen trends **(a)** bewtween 100 and 1000 m, **(b)** at 65°E and **(c)** at 90°E. Dashed gray lines mark **(a)** 65°E and 90°E, and **(b,c)** 100 and 1000 m. Solid black contours represent 20, 60 and 150 μmol/kg oxygen. Dashed black contours highlight salinity signatures of Subtropical Underwater (STUW) and Intermediate Water (IW), with Central Water (CW) and Mode Water (MW) between. Main features of dissolved oxygen trends numbered, with effect on OMZ volume indicated by purple and red arrows. Results are stippled where less than 75 % (6/8) of models agree on sign of trend.

spread is tighter for trends in OMZ20, and so the multi-model mean signal of volume trends emerges in the ensemble earlier for
OMZ20 than for OMZ60 (Fig. 3b,c). About half of the OMZ20 contraction occurs in the Arabian Sea (Fig. S8). Meanwhile, the volume of OMZ150 falls between the regimes of robust expansion and contraction and experiences near-zero volume changes in the multi-model mean by 2100. Tracking the evolution of the OMZ150, we see that ESMs project an expansion over the historical period and first half of the twenty-first century before disagreeing on the trajectory in the latter half of the century, with half of the ESMs projecting a contraction by 2100 (GFDL-CM4, MPI-ESM1-2-LR, UKESM1-0-LL, MIROC-ES2L) and
the other half projecting an expansion (GFDL-CM4, MPI-ESM1-2-HR, NorESM2-LM, NorESM2-MM; Fig. 3d). Lastly, the volume of waters with less than 200 μmol/kg evolves similarly to the volume of the OMZ150 over the historical period, but then continues to follow a trajectory of robust expansion over the twenty-first century (Fig. 3e).

### 3.3  Regional contrasts in dissolved oxygen trends reshape the Indian Ocean OMZ

Projected trends in dissolved oxygen are highly variable in space, both horizontally and vertically, leading to regional con-
traction, expansion and redistribution of the OMZ volume (Fig. 4). The multi-model mean projects a weak increase in oxygen in the OMZ cores of the Arabian Sea and Bay of Bengal and a decline in oxygen in the relatively well oxygenated water

masses entering via the Southern Pathway (Fig. 4). Strongest oxygen changes are, however, found at intermediate oxygen levels. We focus on the three major features that influence the OMZ volume at these intermediate levels (labeled by numbers in Fig. 4): the deoxygenation in the southeastern basin (Feature 1), the oxygenation of the western South Equatorial Current (Feature 2), and the deoxygenation in the northern Arabian Sea (Feature 3). Feature 1 coincides with the flow of Southern Pathway Waters in the subtropical gyre and Indonesian Throughflow Water entering the Indian Ocean. Indeed, we see two distinct maxima of the deoxygenation along the meridional section at $90°$ E (Fig. 4c): one coinciding with Southern Pathway waters (namely Central, Mode and Intermediate Waters) centered around 500 m depth with deoxygenation rates up to $35 \, \mu \mathrm{mol/kg/century}$, and one coinciding with Indonesian Throughflow waters at around 250 m depth with deoxygenation rates up to $50 \, \mu \mathrm{mol/kg/century}$. Feature 2 is located where Southern Pathway Waters and Indonesian Throughflow waters mix within the South Equatorial Current. Feature 2 has a maximum oxygenation trend in the South Equatorial Current between 100 and 500 m, where waters oxygenate at a rate of up to about $50 \, \mu \mathrm{mol/kg/century}$, and a secondary maximum in the North Equatorial Current, with oxygenation rates up to $40 \, \mu \mathrm{mol/kg/century}$ (Fig. 4a-b). Together, Features 1 and 2 form an oxygenation-deoxygenation dipole along the path of the South Equatorial Current. Lastly, Feature 3 is a deoxygenation at a rate of up to about $60 \, \mu \mathrm{mol/kg/century}$ in the northern Arabian Sea between about 100 and 300 m depth, a region where the Persian Gulf Outflow (observed down to about 400 m depth) and winter convection (observed mixed layers down to about 100 m depth) dominate ventilation (Fig. 4a-b).

The three features of dissolved oxygen trends are simulated by all individual ESMs in the ensemble (Fig. S9,S10), leading to a robust pattern in the multi-model mean (more than 75 % of models agree on sign of trend; Fig. 4). However, there are some notable differences in their amplitude and extent across the ensemble. Here, we leverage comparisons between models that share the same architecture, but differ either in their horizontal resolution or the complexity of their biogeochemical module, to illustrate the sensitivity of the oxygen response to the model characteristics. For example, GFDL-CM4 and GFDL-ESM4 have the same dynamical ocean model but differ in both oceanic and atmospheric resolution (GFDL-CM4 has higher resolution) and biogeochemical components (GFDL-ESM4 has more complex representation of biogeochemistry) (Dunne et al., 2020; Held et al., 2019). Despite very similar historical representations of oxygen in the Indian Ocean, GFDL-CM4 and GFDL-ESM4 simulate different strengths of oxygenation in the SEC (Feature 2) and deoxygenation in the Arabian Sea (Feature 3), with GFDL-CM4 projecting more pronounced features in both regions (Fig. S10). Meanwhile, NorESM2-LM and NorESM2-MM, which differ only in the horizontal resolution of their atmospheric components (Seland et al., 2020), project similar trends for oxygenation in the SEC but differ on the strength of the deoxygenation in the Arabian Sea, with weaker deoxygenation simulated in the higher atmospheric resolution version (NorESM2-MM; Fig. S10). Finally, CanESM5 and CanESM5-CanOE only differ in the complexity of their biogeochemical model (Christian et al., 2022). Both CanESM models were removed from our multi-model mean due to the absence of suboxia in the Arabian Sea, but we use them here to illustrate the sensitivity to the biogeochemical complexity. CanESM5-CanOE, which has a more complex representation of biogeochemistry, projects weaker oxygenation in the SEC and stronger deoxygenation in the Arabian Sea than the simpler CanESM5 model. These results suggest that the sign of oxygen change is robust across the ensemble, but that both model resolution (atmospheric and oceanic) and biogeochemical complexity influence the amplitude of the change.

These three features ultimately explain the contraction, expansion, and redistribution of the OMZ volume at thermocline depths across oxygen thresholds. The deoxygenation of the southeastern basin (Feature 1) is the primary driver of the volume expansion at high oxygen thresholds ($> 180\,\mu\mathrm{mol/kg}$, Fig. 3b). The oxygenation-deoxygenation dipole (from Features 1 and 2) redistributes the OMZ volume from west to east and explains the weak volume changes found at intermediate oxygen thresholds (between 85 and $180\,\mu\mathrm{mol/kg}$) including the volume of the OMZ150 (redistribution regime, Fig. 3b). The oxygenation in the South Equatorial Current, and in turn the western boundary current waters (Feature 2), contributes to the OMZ contraction at low oxygen thresholds including the volume of OMZ60 (contraction regime, Fig. 3b), although the net change in volume is relatively small due to the compensating expansion in the northern Arabian Sea which also affects the low oxygen thresholds (Feature 3).

## 3.4 Dissolved oxygen trends are dominated by changes in ventilation

To identify which processes control projected dissolved oxygen trends, and in turn the reshaping of the Indian Ocean OMZ, we separate dissolved oxygen trends into thermal and non-thermal components (Fig. 5). Thermal oxygen trends relate to changes in oxygen saturation concentrations, while non-thermal oxygen trends arise from changes in physical ocean circulation and biological respiration rates (Sect. 2.2.2). Thermal and non-thermal effects tend to influence different water masses: thermal trends are strongest in surface waters (above 100 m), while non-thermal trends are strongest at thermocline depths (100 to 1000 m), but are also responsible for deoxygenation in the deep ocean (below 1000 m) (Fig. 5). Thermal oxygen changes drive relatively uniform deoxygenation in the surface ocean of about 5 to $15\,\mu\mathrm{mol/kg/century}$. Thermal effects are only significant below 100 m along the pathways of Subtropical Underwater (Fig. 5b) and Indonesian Throughflow Water (Fig. 5a,c). The non-thermal component largely explains the strong patterns of oxygenation and deoxygenation discussed above, including deoxygenation in the southeast, oxygenation in the SEC, and deoxygenation in the northern Arabian Sea (Features 1 to 3; Fig. 4 and 5d-f).

The ideal age of seawater, or the average time since water has been exposed to the surface, is a direct measure of ventilation timescale; trends in ideal age thus indicate changes in the rate of ventilation from advection and mixing. Trends in ideal age agree with the pattern of non-thermal oxygen trends. The regions of maximum decreased ventilation (i.e., waters get older) correspond spatially with deoxygenation in the southeastern basin (Feature 1) and Arabian Sea (Feature 3), while the region of maximum increased ventilation (i.e., waters get younger) corresponds to the region of oxygenation in the South Equatorial Current (Feature 2; Fig. 5ghi, Fig. S11). This suggests that changes in physical ventilation pathways are the main control of features of dissolved oxygen change, and ultimately the reshaping of OMZ volume, in the Indian Ocean.

## 3.5 Shifts in water mass composition control oxygenation in South Equatorial Current

In this and the following section, we attribute the significant features of projected oxygen trends in the Indian Ocean thermocline to specific changes in ventilation pathways. Perhaps the most remarkable aspect of projected dissolved oxygen changes in the Indian Ocean under SSP5-8.5 forcing is the oxygenation-deoxygenation dipole along the South Equatorial Current (SEC) (Features 1 and 2). The two major ventilation pathways in this region are the Southern Pathway Waters and Indonesian

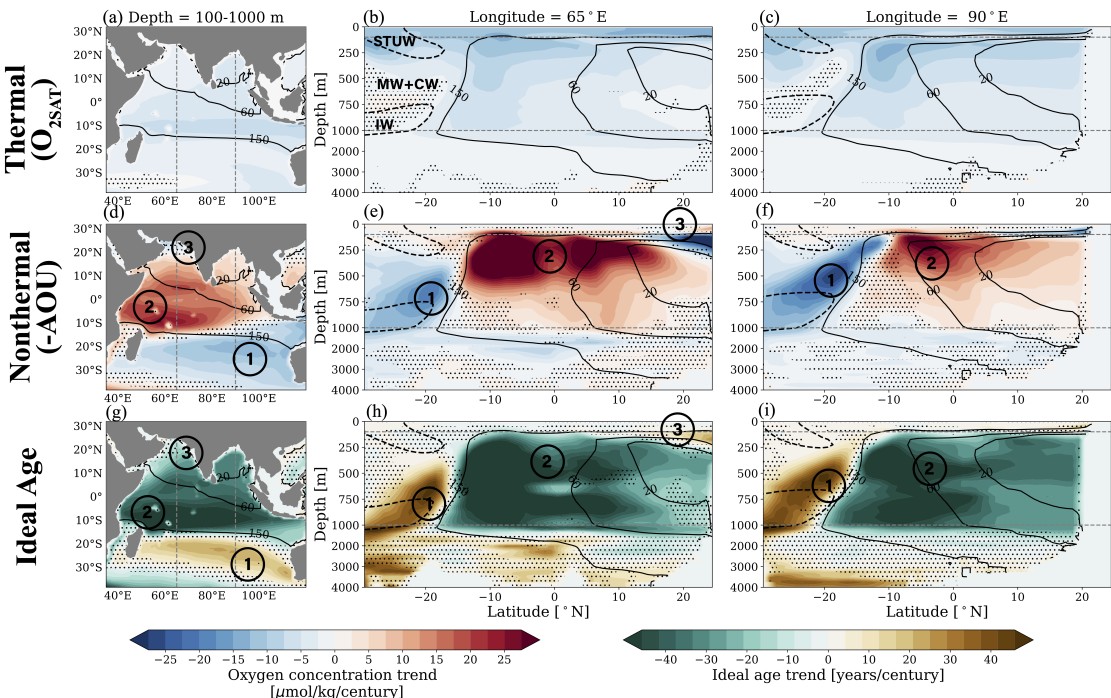

**Figure 5.** Thermal and non-thermal components of multi-model mean (MMM) dissolved oxygen trends under SSP5-8.5 scenario forcing (2015-2100). Thermal oxygen ($O_{2SAT}$) trends **(a)** between 100 and 1000 m, **(b)** at 65°E and **(c)** at 90°E. Non-thermal oxygen (-AOU) trends **(d)** between 100 and 1000 m, **(e)** at 65°E and **(f)** at 90°E. Multi-model mean (MMM) ideal age trends **(g)** between 100 and 1000 m, **(h)** at 65°E and **(i)** at 90°E. Dashed gray lines mark **(a,d)** 65°E and 90°E, and **(b,c,e,f,h,i)** 100 and 1000 m. Solid black contours represent 20, 60 and 150 $\mu$mol/kg oxygen. Dashed black contours represent salinity contours highlighting Subtropical Underwater (STUW) and Intermediate Water (IW), with Central Water (CW) and Mode Water (MW) between. Main features of dissolved oxygen trends numbered. Results are stippled where less than 75 % (6/8) of models agree on sign of trend.

Throughflow Water. The transport of these two pathways is projected to decline between 100 and 1000 m over the twenty-first century (Fig. S12). The transport of Southern Pathway Water across 30° S is projected to decline by 4.4±3.2 Sv/century (from 53±2 Sv) and the transport of Indonesian Throughflow Water by 5.1±2.0 Sv/century from (15±3 Sv). The weakening of these transport pathways can explain the deoxygenation in the southeastern Indian Ocean where each pathway acts as a single dominant source (thus accounting for Feature 1). Curiously, however, this weakened transport seems in apparent con-

tradiction with the increased ventilation and oxygenation of the western SEC (Fig. 4). We hypothesize that this oxygenation must arise from a shift in the relative contributions between these source waters in the western SEC where they co-dominate, favoring relatively young and oxygen-rich waters from the Southern Pathway over relatively old and oxygen-poor waters from the Indonesian Throughflow. To test this hypothesis, we perform an mixture model analysis (Tomczak and Large, 1989) in the tropical Indian Ocean for both a historical mean state (1950-2015) and an end-of-twenty-first-century mean state (climatolog-

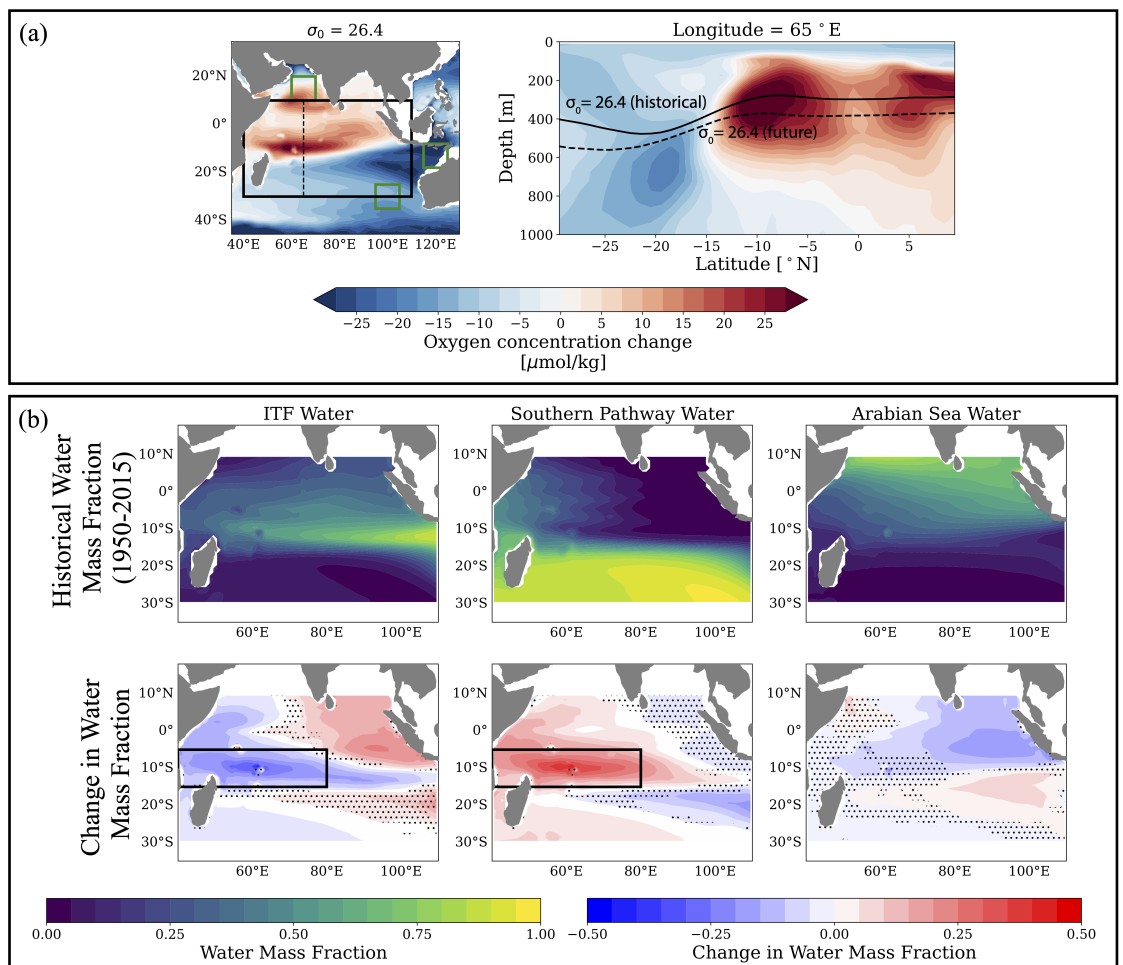

**Figure 6.** Multi-model mean (MMM) results of mixture model analysis in the South Equatorial Current (SEC). **(a)** Oxygen changes along the $\sigma_0 = 26.4$ surface and the historical (solid) and end-of-century (dashed) depth of $\sigma_0 = 26.4$ indicated over MMM oxygen changes in depth space at 65 °E. Green boxes mark source water regions and solid black box represents region for which water mass fractions are solved. (b) Historical distribution of Indonesian Throughflow, Southern Pathway, and Arabian Sea Water fractions, and changes in water mass fractions between historical and end-of-century states. Historical states calculated as 1950-2015 mean, and end-of-century states calculated as historical mean plus linear trends over 2015-2100. Solid black boxes in (b) show region of western SEC. Stippling where less than 75 % (6/8) of models agree on sign of change.

ical 2100) at a potential density layer within the thermocline, nominally $\sigma_0 = 26.4$ (Sect. 2.2.3). This potential density layer intersects the dipole of oxygen change (Features 1 and 2) in the South Equatorial Current (Fig. 6a). We solve for the fractions of three source waters: Indonesian Throughflow Water, Southern Pathway Water and Arabian Sea Water (oxygen minimum water; Sect. 2.2.3). We find that the mixture model analysis is able to reconstruct historical and future oxygen, temperature and

salinity in this region with composites of source water types (see Fig. S13 for a comparison of simulated and reconstructed states in the Feature 2 region).

In the multi-model mean historical state, the mixture model analysis highlights Indonesian Throughflow Water originating from the main straits of the Indonesian Throughflow and penetrating across the basin along $10°$ S, and Southern Pathway Water residing primarily within the southern gyre until it joins the western boundary current (Fig. 6b; in agreement with observed distribution from Tomczak and Large, 1989). Specifically, the mixture model suggests that waters in the region of Feature 2 in the western SEC are composed of about 40 % Indonesian Throughflow Water, 30 % Southern Pathway Water and 30 % Arabian Sea Water in the multi-model mean (average composition between 40-80° E and 5-15° S, Fig. 6b). Under SSP5-8.5 forcing, there is a significant shift in water mass fractions in the western SEC away from Indonesian Throughflow Water in favor of Southern Pathway Water between the historical and future states. By the end of the century, the composition of this region is projected to shift to about 20 % Indonesian Throughflow water and 55 % Southern Pathway Water (changes of -.2 and +.25 water mass fractions for Indonesian Throughflow and Southern Pathway Waters, respectively), while the contribution of Arabian Sea Water stays relatively constant (Fig. 6b).

In all ESMs except one (MPI-ESM1-2-LR), the simulated oxygenation in the western SEC is well approximated by the change in oxygen supply associated with the shift in source water composition in the mixture model (simulated $\Delta O_2$ and oxygen supply reconstructed from water mass compositions, $\Delta O_{2,\text{supply}}$, fall within $\pm 5 \, \mu\text{mol/kg}$ of the 1-to-1 line in Fig. 7a). This suggests that changes in remineralization of oxygen might have a relatively low influence compared to changes in transport (Sect. 2.2.3). Furthermore, nearly all of the changes in oxygen can be accounted for by the changing supply from the Indonesian Throughflow and Southern Pathway ($\Delta O_2$ and $\Delta O_{2,\text{supply}}$ from ITF and SPW alone fall near the 1-to-1 line in 7b). That is, the oxygenation in the western SEC can be accounted for by a shift away from oxygen-poor Indonesian Throughflow Water in favor of oxygen-rich Southern Pathway Water. We note that MPI-ESM1-2-LR is the only ESM in the ensemble which behaves qualitatively differently than the multi-model mean, projecting an increased influence of Indonesian Throughflow Water, and weakened influences of Southern Pathway and Arabian Sea Waters in this region (not shown). MPI-ESM1-2-LR is also the only model that requires significant contributions from remineralization to explain the simulated changes in oxygen. Specifically, the mixture model requires significantly weaker remineralization in the future state than in the historical state to match the simulated oxygen fields (larger oxygenation $\Delta O_2$ than reconstructed from mixture model $\Delta O_{2,\text{supply}}$ for MPI-ESM1-2-LR in Fig. 7a).

The modified water mass composition in the western SEC is then propagated across the equator by the western boundary current along the Somali coast and then east into the basin interior by the North Equatorial Current, increasing ventilation in the northern basin and accounting for the secondary oxygenation maximum north of the equator (Fig. 6b). A previous study found that ESMs projected a strengthening of the cross-equatorial transport and North Equatorial Current with warming (Sharma et al., 2023), which may in turn strengthen the northern oxygenation maximum. We note that despite the movement of the $\sigma_0 = 26.4$ surface between the historical and end-of-century states, the results here are not an artifact of this isopycnal displacement (Fig. S14)

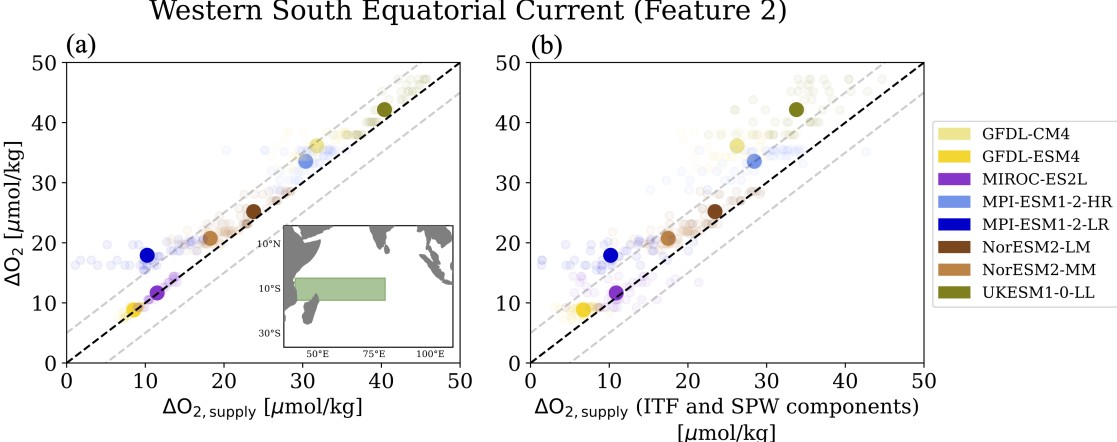

**Figure 7.** Dissolved oxygen changes computed from water mass composition changes, $\Delta O_{2,supply}$, versus total simulated dissolved oxygen changes in the western South Equatorial Current (indicated by inset in panel **(a)**). **(a)** $\Delta O_{2,supply}$ from all source water changes. **(b)** $\Delta O_{2,supply}$ only from contributions of Indonesian Throughflow and Southern Pathway Waters. Solid markers are mean of all samples (faded markers).

### 3.6 Shoaling marginal sea outflows shift ventilation in the Arabian Sea

We perform a similar analysis as above to assess the role of changing ventilation pathways, namely marginal sea outflows, to
the deoxygenation projected in the northern Arabian Sea (Feature 3). Unlike pathways in the southern Indian Ocean, the overall strength of marginal sea outflow transport remains steady under SSP5-8.5 forcing (Fig. S12). Neither the Persian Gulf outflow nor the Red Sea outflow show any robust trends in volume transport across the ESM ensemble. We hypothesize that, although overall transport from marginal seas does not change, the buoyancy of outflow plumes increases with the rapid heating of marginal seas, which can shoal the outflow plumes and reduce ventilation of the OMZ at thermocline depths. This hypothesis
is motivated by the findings of Lachkar et al. (2019), who find that this mechanism can have a strong influence on the OMZ in the Arabian Sea. To test this, we perform an mixture model analysis along the nominal potential density layer $\sigma_0 = 25.7$, which corresponds to the region of deoxygenation in the Arabian Sea (Feature 3; Fig. 8a). We define three source waters for the region: Persian Gulf Water, Red Sea Outflow Water and Arabian Sea Water (Sect. 2.2.3). We find that the mixture model analysis is able to reconstruct historical and future oxygen, potential temperature and salinity from the three source water types
in the northern Arabian Sea region (see Fig. S15 for a comparison of simulated and reconstructed states in the region of Feature 3).

In the multi-model mean historical state (1950-2015), Persian Gulf Water spreads from the Gulf of Oman across the northern Arabian Sea and Red Sea Water spreads from the Gulf of Aden, while Arabian Sea Water acts as the ambient water in the basin (Fig. 8b). In the region of Feature 3 (between 60-70° E and 20-25° N), Persian Gulf Water accounts for about 65 % of the
water mass in the multi-model mean historical state. Concurrent with the region of deoxygenation in the northern Arabian Sea,

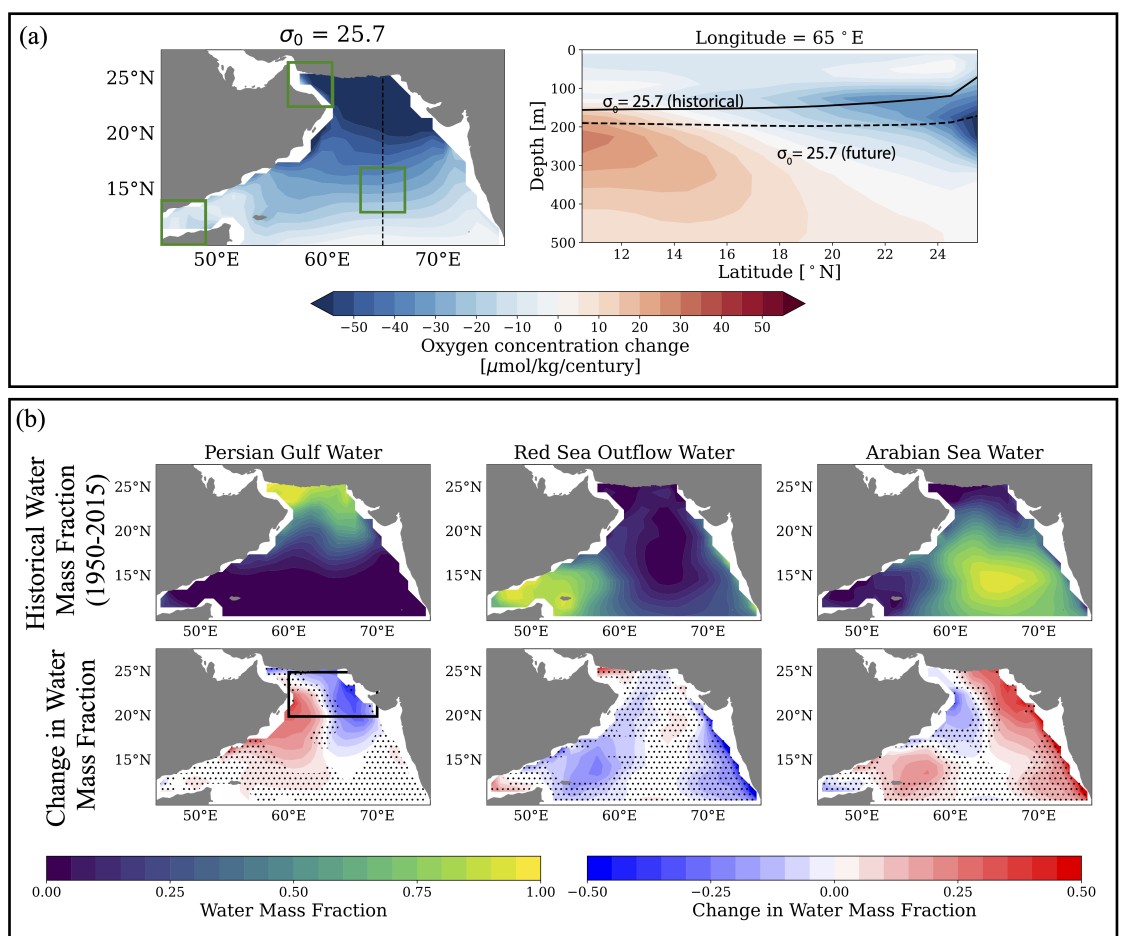

**Figure 8.** Multi-model mean (MMM) results of mixture model analysis in the Arabian Sea. **(a)** Oxygen changes along the $\sigma_0 = 25.7$ surface and the historical (solid) and end-of-century (dashed) depth of $\sigma_0 = 25.7$ indicated over MMM oxygen changes in depth space at 65 °E. Green boxes mark source water regions. (b) Historical distribution of Persian Gulf, Red Sea Outflow, and Arabian Sea Water fractions, and changes in water mass fractions between historical and end-of-century states. Historical states calculated as 1950-2015 mean, and end-of-century states calculated as historical mean plus linear trends over 2015-2100. Solid black box in (b) show region of northern Arabian Sea over which we report changes. Stippling where less than 75 % (6/8) of models agree on sign of change

there is a decrease in the presence of Persian Gulf Water in the end-of-century mean, which gets replaced by ambient Arabian Sea Water. Averaged over the region of Feature 3, the contribution of Persian Gulf Water decreases to 55 % by the end of the century (decrease in water mass fraction of about .1; Fig. 8b). This is also associated with a shift in Persian Gulf Water westward. Further south, there is a decrease in ventilation from Red Sea Outflow Water along $\sigma_0 = 25.7$, which may contribute

to deoxygenation off of the Omani coast (Fig. 8b).

**Figure 9.** Dissolved oxygen changes computed from water mass composition changes, $\Delta O_{2,\text{supply}}$, versus total simulated dissolved oxygen changes in the northern Arabian Sea (indicated by inset in panel **(a)**). **(a)** $\Delta O_{2,\text{supply}}$ from all source water changes. **(b)** $\Delta O_{2,\text{supply}}$ only from contributions of Indonesian Throughflow and Southern Pathway Waters. Solid markers are mean of all samples (faded markers).

In five out of the seven ESMs, the simulated deoxygenation in the northern Arabian Sea is well approximated by the change in oxygen supply reconstructed from the shift in water masses in the mixture model ($\Delta O_2$ and $\Delta O_{2,\text{supply}}$ fall within $\pm 5\,\mu\text{mol/kg}$ of the 1-to-1 line in Fig. 9a). We find that the majority of this deoxygenation in the region can be accounted for by changes in oxygen supply by the Persian Gulf ($\Delta O_2$ and $\Delta O_{2,\text{supply}}$ from Persian Gulf Water alone fall close to the 1-to-1 line in Fig. 9b).

We note that two ESMs (MPI-ESM1-2-HR and MPI-ESM1-2-LR) exhibit weaker $\Delta O_{2,\text{supply}}$ than $\Delta O_2$ (ESMs fall below the 1-to-1 line in Fig. 9a). While this may suggest that remineralization increases between the historical and future states in these ESMs, this is unlikely given projected changes in export of organic carbon (see next section). Rather, it is likely that the region is poorly described by the source water types given to the mixture model in these two ESMs.

We note that for an alternative analysis where the future state is evaluated at a potential density layer which aligns approxi-

mately with the historical depth of $\sigma_0 = 25.7$ (i.e., aliasing from isopycnal displacement is removed; Sect. 2.2.3; Fig. S16), we see greater agreement between the pattern of deoxygenation and decreases in Persian Gulf Water fraction.

## 3.7 Contributions of biological and stratification changes to oxygen changes

The mixture model analysis suggests that nearly all of the simulated oxygen changes in the western SEC and northern Arabian Sea can be accounted for by shifts in water mass compositions driven by changes in advective ventilation pathways (Fig. 7, 9).

However, the residual oxygen changes that the mixture model analysis does not capture, while small for most ESMs, tend to be systematically biased positive for the western SEC and negative for the northern Arabian Sea. This suggests the influence of additional processes not accounted for in the reconstruction of oxygen supply from the mixture model analysis, in particular the increase in subsurface stratification expected with global warming that can influence the vertical mixing between different

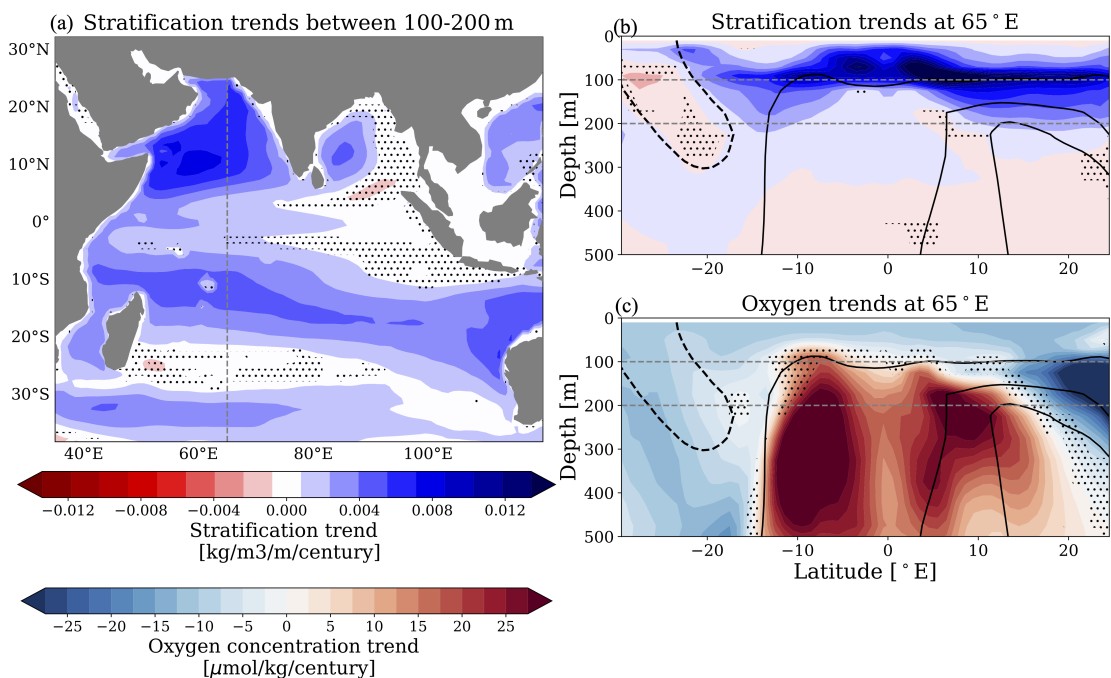

**Figure 10.** Multi-model mean (MMM) stratification trends under SSP5-8.5 scenario forcing (2015-2100). Stratification trends **(a)** between 100 and 200 m and **(b)** at 65°E. **(c)** Dissolved oxygen trends at 65°E. Dashed gray lines mark **(a)** 65°E and 90°E, and **(b,c)** 100 and 200 m. Solid black contours represent 20, 60 and 150 µmol/kg oxygen. Dashed black contours highlight salinity signatures of Subtropical Underwater (STUW). Results are stippled where less than 75 % (6/8) of models agree on sign of trend.

water masses, and changes in biological oxygen consumption along advective pathways once a water parcel has left its source water region.

We examine trends in stratification (for the eight ESMs) and the export of organic carbon at 100 m (available for seven out of eight ESMs) under SSP5-8.5 forcing. Strongest stratification increases are generally confined to the upper 100 m of the water column, but in both the Arabian Sea and SEC regions, the multi-model mean simulates significant stratification increases between 100 and 200 m (about 0.005 $\mathrm{kg/m^3/m/century}$, Fig. 10). These stratification increases are simulated consistently across the ESM ensemble for both regions (Fig. S17). The oxygenation in the SEC region occurs mostly below 200 m, where stratification does not show strong changes, suggesting that changes in vertical mixing likely play a relatively minor role in this region (Fig. 10b,c). In contrast, the deoxygenation in the northern Arabian Sea is largely collocated with the increase in stratification simulated between 100 and 200 m, suggesting that reduced vertical mixing of oxygenated surface waters downward contributes to the deoxygenation projected in this region. Increased subsurface stratification tends to limit the vertical mixing of oxygenated surface waters downward, but also limits the mixing of subsurface nutrients into the surface, which in turn limits primary productivity, the export of organic matter at depth, and potentially reduces the consumption of

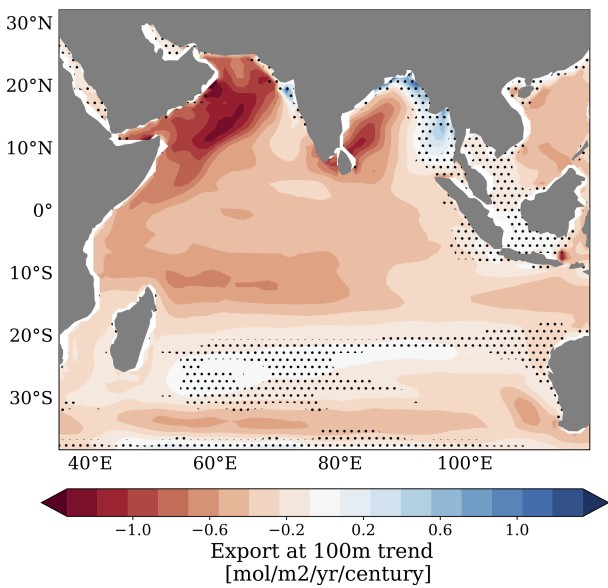

**Figure 11.** Multi-model mean (MMM) export of organic carbon at 100 m trends under SSP5-8.5 scenario forcing (2015-2100). Results are stippled where less than 75 % (6/8) of models agree on sign of trend.

oxygen in the subsurface. The export of organic carbon consistently declines under SSP5-8.5 forcing over the Indian Ocean in the multi-model mean, with local maximum declines in the western SEC, the Bay of Bengal and the Arabian Sea (Fig. 11 and Fig. S18 for individual ESMs), suggesting a decline in the consumption of oxygen along ventilation pathways.

The projected decline in export and increase in stratification have opposing effects on oxygen changes, and are therefore at least partially offsetting each other. The CMIP6 archive does not provide the necessary data to quantify the contributions of these two factors on oxygen changes. However, the signs of the residual oxygen changes that remain unexplained by the mixture model analysis can provide some qualitative information on the relative contributions of these two processes. Positive residuals (i.e. stronger oxygenation simulated in ESMs than reconstructed by the mixture model analysis; Fig. 7) suggests
that the decline in export dominates over the increase in stratification in the western SEC. In contrast, negative residuals (i.e., stronger deoxygenation simulated in ESMs than reconstructed by the mixture model analysis, Fig. 9) suggest that the influence of increased stratification exceeds the influence of reduced export in the northern Arabian Sea.

## 4   Discussion

We examine the changes in dissolved oxygen and OMZ volume in response to the high-emissions scenario forcing (SSP5-8.5)
in the Indian Ocean using an ensemble of eight CMIP6 Earth system models. In the following, we discuss the three regimes that characterize the OMZ response to warming in the Indian Ocean thermocline (upper 1000 m). We compare this response to that previously investigated for the Pacific Ocean OMZ and to the unstudied response for the Atlantic OMZ, and contrast the OMZ

changes projected in the thermocline to the changes projected in the deep ocean (Fig. 12). We also revise the existing "single-pipe" and "mixing-network" ventilation framework to interpret the OMZ response of the Indian Ocean, discuss observational constraints on ventilation pathways, and consider caveats in the model ensemble projections.

## 4.1 Three-regime response of the oxygen minimum zone

The response of oxygen and the OMZ in the Indian Ocean thermocline (upper 1000 m) falls into three regimes (Fig. 3a and 12a): a contraction of volume delimited by low oxygen thresholds ($<85\,\mu\mathrm{mol/kg}$), an expansion of volume delimited by high oxygen thresholds ($>180\,\mu\mathrm{mol/kg}$), and a redistribution of the volume at intermediate oxygen thresholds (85 to $180\,\mu\mathrm{mol/kg}$). This three-regime response is similar to the OMZ response described in the Pacific Ocean by Busecke et al. (2022) (Fig. 12b). A striking difference between the two basins, however, is the widespread oxygen gain projected in the Indian Ocean. Indeed, ESMs robustly project an oxygenation of the OMZ cores in the Arabian Sea and Bay of Bengal, similar to the Pacific OMZ core, but also an oxygenation of the southwest Indian Ocean along the path of the South Equatorial Current and North Equatorial Current. As a result, the OMZ contracts for thresholds as high as $85\,\mu\mathrm{mol/kg}$ (including the volume of hypoxic waters commonly defined by $60\,\mu\mathrm{mol/kg}$) in the Indian Ocean, whereas the contraction is restricted to the very low oxygen levels found in the OMZ core in the Pacific Ocean (typically volume with oxygen less than 10 to $20\,\mu\mathrm{mol/kg}$; Fig. 12a,b). The oxygen gain in the southwest Indian Ocean is associated with an oxygen loss in the southeast. This dipole of southwest oxygenation and southeast deoxygenation along the path of the South Equatorial Current explains the redistribution regime and near zero change of volumes delimited by intermediate oxygen levels (85 to $180\,\mu\mathrm{mol/kg}$; Fig. 12b). Finally, upstream of the South Equatorial Current, the drastic decline in oxygen supplied by the water masses that enter the thermocline through the South Indian Gyre and Indonesian Throughflow explains the expansion of volumes defined by high oxygen levels ($>180\,\mu\mathrm{mol/kg}$; Fig. 12a). The slowdown of the Southern Pathway is likely driven either by a weakening of Indian Central Water or Subantarctic Mode Water formation, but we do not disentangle these two effects in this study.

## 4.2 Beyond "single-pipe" and "mixing-network": cases for "two-pipe" and "moving pipe" systems

The projected oxygenation outside of the Indian OMZ core, or shadow zone, calls for a revision of the "single-pipe" and "mixing-network" ventilation framework used in the Pacific Ocean (Gnanadesikan et al., 2007). In the original framework, increased ventilation and oxygenation with weakened circulation was only reconcilable in shadow zones where there are no direct advective ventilation pathways and a mixing-network of multiple sources controls ventilation. In both the Pacific and the Indian Oceans, the oxygenation of the OMZ core (which is associated with increased ventilation and reduced ideal age) can indeed be interpreted as a shift towards a stronger contribution from younger, oxygen-rich upper ocean waters using the mixing-network model (Bryan et al., 2006; Gnanadesikan et al., 2007, 2012; Takano et al., 2018; Busecke et al., 2022). In the Pacific, this effect has been attributed to a slowdown of Deep Waters upwelling to thermocline depths, and there is evidence of this same effect in the Indian Ocean basin. In the ESM ensemble used in this study, upwelling of Deep Waters across 1000 m weakens at a rate of $2.0\pm0.7\,\mathrm{Sv/century}$ in the Indian Ocean, even reversing sign to be net downwelling (under SSP5-8.5 forcing in this ESM ensemble, see Fig. S12).

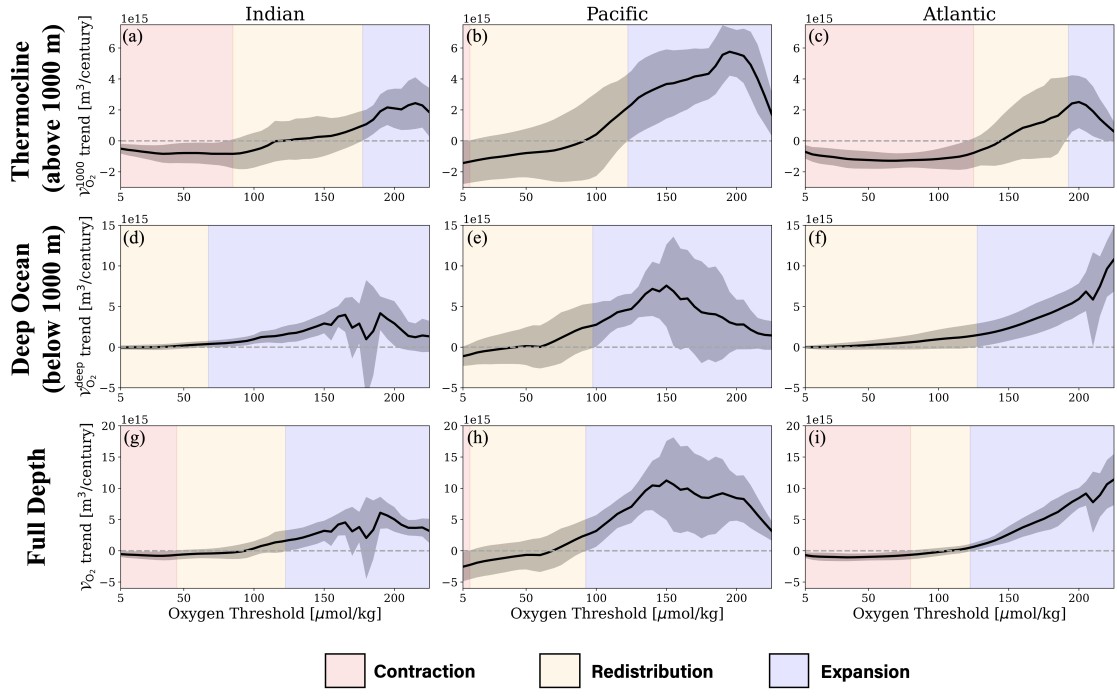

**Figure 12.** Multi-model mean thermocline OMZ volume trends (above 1000 m) under SSP5-8.5 scenario forcing (2015-2100) as a function of oxygen threshold for the tropical (30° S - 30° N) **(a)** Indian, **(b)** Pacific and **(c)** Atlantic Oceans. Deep ocean (below 1000 m) OMZ volume trends for the tropical **(d)** Indian, **(e)** Pacific and **(c)** Atlantic Oceans. Full depth OMZ volume trends for the tropical **(g)** Indian, **(h)** Pacific and **(i)** Atlantic Oceans. Shading represents one standard deviation of model spread from the mean. Panel a is the same as Fig. 3a

Changes in mixing in the OMZ core are, however, not the primary effect driving oxygenation in the Indian Ocean. In the Southern Indian Ocean, ESMs project an oxygenation and increased ventilation well outside of the shadow zone, within regions of strong advection such as the South Equatorial Current. We show that this oxygenation can be interpreted as a "two-pipe" system, an intermediate case between a "single-pipe" and a "mixing-network". Two sources with different oxygen levels, rather than a network of many sources, are in fact sufficient to have increased oxygenation, even though the transport by both ventilation sources weaken. The unique configuration of the Indonesian Throughflow and Southern Pathway merging into the South Equatorial Current allows for this phenomenon despite intense advection. The ventilation by the Southern Pathway and Indonesian Throughflow both decline in response to warming in the ESM ensemble, but the ventilation by the Indonesian Throughflow declines more strongly, favoring the relative contribution of the better oxygenated Southern Pathway waters.

Another feature of the Indian Ocean OMZ response which differs from its Pacific Ocean counterpart is the importance of ventilation by marginal sea outflows. We find that changes in ventilation from marginal sea outflows may be a significant driver of deoxygenation in the Arabian Sea. The Persian Gulf rapidly warms in response to global warming, increasing stratification and the buoyancy of outflow water from the Persian Gulf. As outflow water shoals, it becomes unable to ventilate the OMZ in

the Arabian Sea, and is instead confined to the upper 100 m where its impact on oxygen becomes marginal (the upper ocean is already well oxygenated; Lachkar et al., 2021). The deeper Red Sea outflow plume also shows evidence of shoaling in the ESM ensemble, but it has a weaker impact on the ventilation of the OMZ because its oxygen content is lower than that of the Persian Gulf (on average about $70 \, \mu \mathrm{mol/kg}$ for Red Sea Outflow Water versus about $200 \, \mu \mathrm{mol/kg}$ for Persian Gulf Water). We can describe this change in the ventilation from marginal seas as a "moving pipe", where oxygen supply may be affected by a displacement of a pathway even if the strength of transport along a that pathway holds steady.

## 4.3 Global trends in tropical oxygen minimum zones

Now extending our scope to all tropical oceans, we find that the three-regime OMZ response identified in the Indian (this study) and Pacific (Busecke et al., 2022) Oceans is also simulated in the Atlantic Ocean (Fig. 12c, S19). The dynamics driving this three-regime response in the Atlantic have not yet been explored in detail, but it is likely that the even more prominent contraction regime found for the Atlantic OMZ is connected to changes in ventilation by the Atlantic meridional overturning circulation, which is projected to weaken in ESMs (IPCC et al., 2013; Bakker et al., 2016). This would lead to a favoring of midlatitude waters from the south, as South Atlantic Mode and Intermediate water subduction rates are projected to hold steady and ventilate lighter densities (Goes et al., 2008; Downes et al., 2009). Looking beyond OMZ volume changes in the ocean thermocline (above 1000 m) to the deep ocean response (below 1000 m), we see that deoxygenation and OMZ expansion in the deep ocean are more ubiquitous than in the thermocline (Fig. 12d-f). Deoxygenation in the deep ocean is likely due to a slowdown of Deep Water transports as the global thermohaline circulation weakens with warming (e.g. Bakker et al., 2016). When integrating the thermocline and deep ocean components of the OMZ response to consider the full depth of the water column (similar to what was done for the Pacific Ocean in Busecke et al., 2022), we find that the thermocline response dominates the OMZ contraction at low oxygen thresholds while the deep ocean response dominates the expansion at high thresholds (Fig. 12g-i).

## 4.4 Observational constraints, limitations and caveats

Projected oxygen changes, and thus OMZ volume changes are subject to uncertainties, in particular the magnitude of the oxygenation/deoxygenation dipole along the South Equatorial Current and the deoxygenation in the northern Arabian Sea. The slowdown of the Southern Pathway and Indonesian Throughflow transports is a robust and well studied feature of ESM projections (Downes et al., 2009; Sen Gupta et al., 2016; Feng et al., 2017; Stellema et al., 2019). The pattern of multi-model mean oxygen trends (Fig. 4) is not sensitive to our model selection protocol, as the three features are also simulated by the six ESMs excluded from the ensemble (Fig. S8). However, when all 14 available ESMs are included in the multi-model mean, we find that the deoxygenation in Features 1 and 3 are enhanced, while the oxygenation in Feature 3 is weakened (Fig. S9).

Historical observations support the slowdown and deoxygenation of the Southern Pathway (Helm et al., 2011; Kobayashi et al., 2012; Ito et al., 2017; McMonigal et al., 2022). A secular weakening of the Indonesian Throughflow has not yet been observed, possibly obscured by significant decadal variability (Liu et al., 2015; Feng et al., 2018), but there is evidence for forced trends in the global thermohaline circulation that controls the Indonesian Throughflow transport (Rahmstorf et al.,

2015; Sun and Thompson, 2020) to support the decline in ventilation by this pathway. However, the relative changes in the two pathways, which controls the oxygenation/deoxygenation dipole, is highly uncertain and unconstrained by observations. This dipole pattern has not emerged in current observation-based estimates of historical oxygen trends (Helm et al., 2011; Ito et al., 2017; Schmidtko et al., 2017), though there is some signal of oxygenation near Madagascar in the product of Ito et al. (2017). A detailed study on the time of emergence of oxygen and volume trends identified in this work would require greater availability of large ensembles with biogeochemistry in future generations of CMIP.

Deoxygenation in the northern Arabian Sea has been detected in observations (reviewed by Lachkar et al., 2023). The rapid heating of marginal seas compared to the open ocean and the subsequent increase in stratification and buoyancy of the marginal sea outflows, particularly for the Persian Gulf, is also supported by observations (Al-Yamani et al., 2017; Naqvi, 2021). The projected changes in outflow ventilation are consistent with the findings of ocean model simulation studies, which showed that the warming and shoaling of the Persian Gulf was a major driver of the deoxygenation in the northern Arabian Sea (Lachkar et al., 2019, 2021). Yet, ESMs tend to overestimate the oxygen transport by these marginal sea outflows to the Arabian Sea (e.g. Schmidt et al., 2021, for CMIP5). This partially explains the systematic high oxygen bias in the Arabian Sea in the ensemble of eight CMIP6 ESMs used in this study (Fig. 1), the absence of a suboxia in the five ESMs that were excluded from the ensemble (Table S1), and could influence the magnitude of the projected deoxygenation associated with the vertical displacement of these outflows. In fact, a study by Vallivattathillam et al. (2023), which applies downscaling methods to CMIP5 projections of Arabian Sea oxygen under the high emission RCP8.5 forcing, finds that the deoxygenation simulated in CMIP5 models in the region is not preserved after the bias-corrected downscaling method is applied (averaging between 200 and 700 m). This suggests that ESM simulated mean state biases in the Arabian Sea can have a profound impact on the projected changes.

An additional challenge of the mixture model analysis is to distinguish between changes due to source water types and changes in mixing with surface waters. Specifically in the Arabian Sea, Persian Gulf Water and Arabian Sea surface water both tend to be anomalously warm, saline and well oxygenated. Thus, it is possible that increased stratification and weakened vertical exchange with the surface is interpreted as a decrease in Persian Gulf Water fraction in the mixture model. The presence of Northern Arabian Sea deoxygenation in MIROC-ES2L, an ESM which does not resolve the marginal seas and was thus excluded from the OMP analysis for the region, also suggests that stratification increases may play a first-order role in Northern Arabian Sea deoxygenation in the ESM ensemble. This is also consistent with a previous study by Lachkar et al. (2021), which investigates deoxygenation in the Arabian Sea from 1982 to 2010 using a high-resolution regional ocean model, and attributes about 75 % of ventilation decreases over this period to vertical mixing changes and only 25 % to advective changes. We note that the compensation between the effects of export and stratification changes may in fact be key to allowing the mixture model to perform well in this region.

Further work, including for example a full oxygen budget analysis or lagrangian particle tracking, would be required to better disentangle the effects of marginal sea shoaling and increased stratification in the projections of Arabian Sea oxygen for the ESMs used in this study; however, data for such a study are not currently available. Based on these considerations, the relative effects of marginal sea shoaling and increased stratification in ESM projections of the northern Arabian Sea remain largely uncertain. Ultimately though, the present study supports the conclusions of the recent review by Lachkar et al. (2023):

the Arabian Sea OMZ will be reshaped by changes in stratification and the Persian Gulf outflow driving deoxygenation in the northern subsurface layer, while increased ventilation from the south drives oxygenation of the deeper layers.

## 5    Conclusion

Oxygen minimum zones can be viewed as multiple concentric layers, like onions, with oxygen concentrations decreasing towards their core. Their response to deoxygenation and global warming in Earth system models falls into three regimes which are controlled by the mechanisms that ventilate these different layers: the outer layers of the oxygen minimum zone expand, the inner layers contract, and in-between, oxygen is redistributed leading to near zero changes in volume but reshaping the spatial distribution of the intermediate layers. We show that this contrast between inner layer oxygenation and outer layer

deoxygenation, which was first identified in the Pacific Ocean (e.g., Gnanadesikan et al., 2007; Busecke et al., 2022), applies to all tropical oxygen minimum zones. However, the unique geometry and ventilation pathways of the tropical Indian Ocean (bounded by continent to the north, Indonesian Throughflows and marginal seas) determine what layers of the oxygen minimum zone experience contraction, redistribution and expansion. The Indian Ocean is characterized by a prominent oxygenation, attributed to changes in the ventilation by the Indonesian Throughflow, and consequently contraction and redistribution regimes

that extend to a much larger range of oxygen minimum zone layers than its Pacific counterpart. Furthermore, the rapid warming and shoaling of marginal sea outflows leads to localized deoxygenation in the inner layers of the oxygen minimum zone of the northern Arabian Sea. We identify that the response of the Atlantic oxygen minimum zone also exhibits regimes of contraction, redistribution, and expansion, but further investigations are required to understand the regional dynamics which produce a pronounced contraction response in that basin.

*Code and data availability.*   The processed data files and code to reproduce results presented here are available at https://doi.org/10.5281/zenodo.8342233

## Appendix A:  Mixture Model and Extended Optimum Multiparameter Analysis

The mixture model used in this study is based on extended Optimum Multiparameter (OMP) Analysis (OMP; Karstensen and Tomczak, 1998), which is a method of solving an over-determined linear mixture model which takes into account the conversion of one tracer to another, generally by remineralization. Detailed explanations of the OMP method can be found in

Tomczak and Large (1989) and Shrikumar et al. (2022), but here we give a brief summary below:

First, we define a set of $N$ source water types with properties, $p$. Let $e_p^i$ be the values for property $p$ and $i$-th source water mass. Each sample in the evaluation region has a value for each property, $s_p^j$. The goal is to solve for the fraction of $i$-th source water type in each sample, $x_i^j$. We can define the objective function:

$$\sum_{i=1}^{N} e_p^i x_i^j = s_p^j + \epsilon_p^j \tag{A1}$$

where $\epsilon_p^j$ is a residual to minimize. Using user defined weights, $W$, to assign relative importance to each property, the residual takes the form

$$\epsilon_p^j = W_p \Big( \sum_i (e_p^i - \mu_p) x_i^j + r_B^A \Delta_j^B - (s_p^j - \mu_p) \Big) \tag{A2}$$

where we center the values for each property, $e_p^i$ with their mean, $\mu_p$. The user defined weights play the role of normalizing across fields. We chose to set weight for dissolved oxygen and AOU at 2 % the weight of potential temperature and salinity fields to account for the greater range of values in sources of dissolved oxygen. The results of the analysis are not sensitive to small changes in weight values. The term $r_B^A \Delta_j^B$ allows for the conservation of two properties, $A$ and $B$, which are exchanged via remineralization The cost function is thus defined as the sum of residuals for every property, $p$, as well as a mass conservation term $\epsilon_M^j$ which penalizes deviations from total mass fractions of unity for each sample point:

$$C^j = (\epsilon_M^j)^2 + \sum_p (\epsilon_p^j)^2 \tag{A3}$$

The PyOMPA implementation of this method allows for a hard constraint on mass conservation, rather than minimizing a residual (Shrikumar et al., 2022). However, we do not use this feature, and thus recover the original method of Karstensen and Tomczak (1998).

We note that the mixing model used in this study differs in several ways from the traditional extended OMP. We solve a 'mixing-triangle' of three source water types, rather than four as used in, for example, Tomczak Jr (1981). Adding an extra degree of freedom to track remineralization, we thus have a total of four degrees of freedom. Typically, remineralization is tracked as the conversion of oxygen to phosphate and nitrate using Redfield Ratios. Since we do not have access to full nutrient fields for the ESM experiments used in this study, we simply allow for the conversion of oxygen to AOU with a ratio of $-1$. As constraints, we use potential temperature, salinity, dissolved oxygen, AOU and mass conservation. Since AOU is a function of dissolved oxygen, potential temperature and salinity, we have four unique constraints. Thus, our mixture model is equivalent to a determined set of linear equations, rather than an over-determined set (i.e. 'Multiparameter Analysis', rather than 'Optimum Multiparameter Analysis'; Tomczak Jr, 1981; Tomczak and Large, 1989). However, since AOU must also be included in the calculation, we leverage the least-squares method used in solving the over-determined problem.

*Author contributions.* Conceptualization: SD, LR, JB. Data curation: SD, JB. Formal analysis: SD. Funding acquisition: LR. Investigation: SD, LR. Methodology: SD, LR, JB. Software: SD, JB. Supervision: LR. Visualization: SD. Writing – original draft: SD, LR. Writing – review and editing: SD, LR, JB

*Competing interests.* The authors declare no conflicts of interest relevant to this study.

*Acknowledgements.* S.D. and L.R. have been supported by the NSF CAREER Award Number 2042672 and the Princeton University High Meadows Environmental Institute Grand Challenge Award for this study. J.B. thanks the Gordon and Betty Moore Foundation (Grant 8434). The authors acknowledge the World Climate Research Programme, which, through its Working Group on Coupled Modelling, coordinated and promoted CMIP6, participating climate modeling groups for producing and publishing their model output, the Earth System Grid Federation (ESGF) for archiving the data and providing access, and the multiple funding agencies who support CMIP6 and ESGF. The authors further thank David Luet for his technical support and his help in creating a mirrored subset of the CMIP6 archive on the Princeton HPC, which was used for the final analysis. The authors thank Jasmin G. John and John P. Dunne at GFDL for providing additional Earth system model data not available via ESGF.

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
