# Peer review of "Unique ocean circulation pathways reshape the Indian Ocean oxygen minimum zone with warming"

_EGUsphere, 2023_

## Referee Comment (RC2)

Anand Gnanadesikan

This paper considers drivers of changes in hypoxia in the Indian Ocean, a critical region for artisanal fisheries and one whose behavior under global warming has not been well characterized. The paper finds three regimes of oxygen change, which correspond to three different driving mechanisms for that change. The authors kindly credit me with distinguishing "single-pipe" from a "mixing network" models and describe how this can be used to distinguish the changes in the northern, central, and southern Indian Oceans and how it needs to be updated to do so. Frankly, I think their mapping onto the different regimes is clearer than what I wrote about- this is an elegantly written paper. The argument that there are three separate regimes is generally well made.

I have three comments, that in some ways parallel those made by the other reviewer. Normally, I would give this a "major" revision since they will require some new analysis, not merely clarification of existing analysis. But I think the basic analysis is sound and don't want to suggest "reconsideration" of the paper, so I'll call this minor.

The first comment regards the role of the overflows. There is a lot more to overflows than the volue that they deliver to the ocean. It's been a problem for a long time in models to get the depths of injection of overflow water correct. If I look at the salinity along 70E (Figure R1), it's clear that there's a signal from high-salinity shelf waters that penetrates for hundreds of m. It is unlikely that models, which generally have problems with numerical entrainment, correctly capture either this process or its sensitivity to changes in climate. This is even true for relatively high-resolution models (see Seddigh-Marvasti et al., 2015 for some discussion of this). It would be good to at least evaluate how much of a problem this is, rather than simply accepting the results of the MMM (cross-sections of the watermass fractions might be useful to look at this and

discern whether there are any systematic errors here). This wouldn't need to be an extra figure in the main text but would make a good one in the Supplemental material and would be useful for evaluating whether there are any systematic errors here.

2. The neglect of changes in productivity is understandable, but it I note that it was also picked up by the other reviewer. One way of addressing this is to look at how much of the change in the oxygen can be accounted for by changes in the O2:age slope and how much can be accounted for by changes in the age itself (i.e. to the extent that $AOU = \overline{J_{O2}} * age \rightarrow \Delta AOU = \overline{J_{O2}} * \Delta age + age * \overline{\Delta J_{O2}}$ ) Showing not just the correlation coefficients but the regression coefficients might help with this.

3. Finally, I wanted to see whether this picture seemed to work in my own suite of coarse models reported in Bahl et al., 2019. These models don't have a Red Sea or Persian Gulf, and also fail to generate an oxygen minimum zone in the Northern Arabian Sea (this, incidentally, supports the point of this paper and others that resolving the impacts of such water is important).I show results for two cases with low and high lateral mixing in the figure below. Interestingly, it does seem that the same 3 regimes show up.

[Figure]

(A) $\Delta O_2$ (mmol/m$^3$), $A_{Redi}$=400

(B) $\Delta O_2$ (mmol/m$^3$), $A_{Redi}$=2400

(C) $\Delta PO_4^{Remin}*(-150)$ (mmol/m$^3$), $A_{Redi}$=400

(C) $\Delta PO_4^{Remin}*(-150)$ (mmol/m$^3$), $A_{Redi}$=2400

Since I have a remineralized phosphate tracer in this model I can also directly attribute the changes to changes in accumulated phosphate, and again, this dominates the pattern at low mixing, though somewhat less at high mixing. Ultimately I think this highlights an interesting question of whether the real world Indonesian throughflow acts as a barrier to or enhancer of tracer mixing. It also raises the question of how much of the intermodel variability is due to how this subgridscale mixing is handled.

That the basic framework seems to work in this model suite as well is encouraging and supports the publication of the manuscript.

Note,however, that despite not having marginal seas, we still get a drop in the North. This seems to be driven by a shallowing of mixed layers (echoing something noted by the other reviewer), which reach over 100m on average in the winter in the Northern Arabian Sea in this model, but shallow substantially under global warming. This may be difficult to capture with the watermass analysis alone, as it is not clear (at least to me) that the MMM will necessarily capture the differences between overflow and surface watermasses in this region. It would be worth examining changes in mixed layer depth to see to what extent this plays a role in the more realistic models.

References (which you don't need to add to the paper unless you really want to)

Bahl, A.A., A. Gnanadesikan and M.A. Pradal, Scaling global warming impacts on ocean ecosystems: Results from a suite of Earth System Models , *Frontiers in Marine Science*,7, 698, https://doi.org/10.3389/fmars.2020.00698. 2020

Seddigh Marvasti, S., A. Gnanadesikan J.P. Dunne, A. Bidokhti and S. Ghader, Challenges in simulating spatiotemporal variability of phytoplankton blooms in the Gulf of Oman, *Biogeosciences.* 13, 1049-1069, 2016.

---

## Author Comment (AC1)

**General Response**

We thank the three reviewers for their constructive and insightful comments. The main concerns among the reviewers fall into 3 categories. We give a brief overview of how we propose to address them and include the proposed new figures here in the general response, and then provide a detailed response below.

1. **Role of changes in stratification and productivity in dissolved oxygen projections.** The reviewers ask for an expanded investigation of potential changes in upper ocean stratification and the biological sink on oxygen changes. Specifically, increased stratification could contribute to deoxygenation in the northern Arabian Sea, and decreased productivity and respiration rates could contribute to oxygenation in the South Equatorial Current (SEC). We propose to add a subsection to our *Results* in which we examine trends in upper ocean stratification (Figure GAC1) and the export of organic carbon at 100 m (Figure GAC2). Both the northern Arabian Sea and SEC regions are projected to experience increases in stratification between 100 - 200 m, as well as decreases in export at 100 m, with opposing effects on oxygen. The spatial correlation of these two trend fields is expected, given the relationship between stratification and nutrient supply via vertical mixing. This suggests that the effects of stratification and export trends, at least, partially compensate each other in both of these regions. We propose to discuss this compensation and how it is consistent with the results of our sections 3.5 and 3.6 (using Optimum Multiparameter (OMP) analysis) which suggests that changes in circulation and water mass fractions related to lateral advective pathways can explain nearly all simulated oxygen changes (direct calculations of mixing-driven oxygen changes will be added in the revised manuscript; Figure GAC3). We have also performed additional sensitivity tests with the OMP method (see new Figure GAC3 and new Method details at end of general response) and propose to add a discussion of the robustness and limitation of the OMP method, and the implications for our results (to be added to *Discussion* section 4.4).

2. **Clarification of approach.** One reviewer is concerned with the definition of Oxygen Minimum Zone (OMZ) used in the manuscript which is a continuum covering a broad range of oxygen values extending beyond the commonly used hypoxic thresholds (e.g., 60 umol/kg). We propose to clarify in the *Methods* section 2.2.1 and throughout the text the rationale for using this broad definition: (1) The relevant thresholds for 'low oxygen' environments can vary significantly depending on the application, such as (in order of increasing threshold) denitrification, mortality of marine organisms, and sublethal stresses on marine organisms. Specifically, thresholds of ~150 umol/kg (and even as high as 200 umol/kg) are commonly used as habitat boundaries for large commercial fish species (eg Vaquer-Sunyer and Duarte, 2008); (2) Using a broad continuum of OMZ thresholds enables us to interpret changes in oxygen volume (including the hypoxic volume) within the context of larger-scale forced changes (contraction, redistribution, expansion).

3. **Robustness of findings across individual Earth System Models (ESM)**. The reviewers note that much of the analysis presented in the manuscript is based on the presentation of Multi-Model Means (MMM) of the ESM ensemble, which may mask

compensating biases and different dynamical responses across models and may be sensitive to model selection. We propose to a) expand the discussion of the maps/sections of individual ESMs already included in the Supplementary Information (SI) of the original manuscript, b) add individual ESMs when presenting OMZ volume and OMZ volume trends in the main text (new Figures GAC4, GAC5, GAC6), and c) compare the effects of water mass fraction changes (from OMP analysis) across individual models (Figure GAC3). Overall, we find that most models exhibit the same regional pattern of oxygen trends and regimes of OMZ volume trends as the MMM, with varying amplitudes of features across models. This will also be discussed in Discussion section 4.4.

We are confident that these revisions will address the concerns of the reviewers. Below, we provide specific details and figures for these revisions, and we address additional reviewer comments.

[Figure]

Figure GAC (General Author Comment) 1: Multi-model mean (MMM) stratification trends under SSP5-8.5 scenario forcing (1915-2100). Stratification trends (a) between 100 and 200m, and (b) at 65E. (c) dissolved oxygen trends at 65E. (b,c) Solid black contours represent 20, 60 and 150 µmol/kg oxygen. Dashed black contours highlight salinity signature of Subtropical Underwater (STUW). Results are stippled where less than 75% (6/8) of models agree on sign of trend.

[Figure]

Figure GAC2: Multi-model mean (MMM) export of organic carbon at 100 m trends under SSP5-8.5 scenario forcing (1915-2100). Results are stippled where less than 75% (6/7) of models agree on sign of trend.

[Figure]

Figure GAC3: Changes in dissolved oxygen accounted for by mixing model analysis. Total oxygen changes (simulated by ESMs) versus oxygen changes attributed to shifts in water mass composition in the OMP mixing mode for the (a) western South Equatorial Current (SEC) and (b) northern Arabian Sea. Results in the western SEC averaged between 40-80E, 5-15S. Results in the northern Arabian Sea averaged between 62-70E, 20-26N. Solid markers are the mean of 50 runs (light markers) of the OMP analysis with perturbed source water properties (See end of general response for details on new methods). Black dashed line marks 1-to-1 relationship, gray dashed lines mark 5 umol/kg deviations from 1-to-1.

**New method to be added to Optimum Multiparameter (OMP, section 2.2.4) analysis for sensitivity analysis and used to produce Figure GAC3 above**:

"From the mixing model results, we quantify the change in oxygen accounted for by shifts in water mass composition for key regions: $\Delta O_{mixing} = \sum_i (f_{i,1} O_{i,1} - f_{i,0} O_{i,0})$, where $f_i$ and $O_i$ are the fraction and oxygen concentration of source water type $i$, for future (1) and historical (0) states. Oxygen is allowed to be remineralized, so $\Delta O_{mixing}$ need not approximate the total simulated change in oxygen. To test the sensitivity of the mixing model results to small changes in the definition of source water types and potential density layer, we average 50 realizations of the experiment applying random perturbations to source water locations and density values. Source water locations are perturbed by up to 5 degrees in latitude and longitude for the South Equatorial Current region and 2 degrees for the Arabian Sea. For both locations, the value of the potential density layer is perturbed by up to 0.1."

We note regarding the OMP analysis that we lowered the weight on oxygen in the optimization compared to the submitted manuscript (lowered from 20% the weight of temperature and salinity to 2%) to more appropriately account for the range of oxygen values in the analysis compared to the range and temperature and salinities (see Appendix A of submitted manuscript for technical details of OMP).

[Figure]

Figure GAC4: (**To replace Figure 2 in main text**) OMZ volume taken between 0-1000 m in the (a) Indian Ocean and sub-basins (b) Arabian Sea and (c) Bay of Bengal for the multi-model mean averaged over 1950-2015 (MMM; black) and observed climatology (WOA18; red). Shading represents one standard deviation of the model spread. Individual ESMs shown in colored curves. Southern boundary of Arabian Sea and Bay of Bengal are at 5N.

[Figure]

Figure GAC5: (**To replace Figure 3 in main text**) Multi-model mean thermocline OMZ volume changes (between 0 - 1000 m) under SSP5-8.5 scenario forcing (2015-2100). (a) $V_{o2}^{1000}$ trends as a function of oxygen threshold for the Indian Ocean. The 20, 60 and 150 umol/kg thresholds bounding OMZ20, OMZ60 and OMZ150 are indicated with grey dashed lines. (b) Time series of $V_{o2}^{1000}$ anomaly from 1950-2100 (anomaly referenced to 1950-2015 mean) for OMZ60 (dark blue) and OMZ150 (teal). Shading represents one standard deviation of model spread.

[Figure]

Figure GAC6: (**To be added to supplementary information**) Multi-model mean thermocline OMZ volume changes (between 0 - 1000 m) under SSP5-8.5 scenario forcing (2015-2100). $V_{o2}^{1000}$ trends as a function of oxygen threshold for the (a) Arabian Sea and (b) Bay of Bengal with OMZ20 volumes indicated for each basin. (c) Time series of $V_{o2}^{1000}$ anomaly from 1950-2100 for OMZ20 volumes in the Arabian Sea and Bay of Bengal. Shading represents one standard deviation of model spread.

**Reviewer 1:**

The paper by Ditkovsky et al. investigates the response of the Indian Ocean OMZ to future climate change. To this end, authors analyze an ensemble of Earth system model high-emissions scenario simulations from the Coupled Model Intercomparison Project phase 6 (CMIP6). They show that the Indian Ocean experiences a broad oxygenation in the western tropical region driven by a decrease in waters supplied by the Indonesian Throughflow in favor of waters supplied from the south, typically richer in oxygen. This results in a prominent contraction of the OMZ there. However, the warming of the marginal seas, particularly that of the Persian Gulf, is suggested to cause a localized deoxygenation in the northern Arabian Sea, leading the OMZ to expand there.

**General comments:**

I enjoyed reading this manuscript that I found generally well structured and well written. The question of future changes in OMZs in general and in the Indian Ocean in particular is critical and of very high relevance to the community, as the previously reported trends remain unclear and tainted with important uncertainties. The authors did an excellent job at summarizing and analyzing the O2 changes simulated by the CMIP6 models in the Indian Ocean for different classes of O2. However, I have a few concerns relative to the presentation and the interpretation of the results that need to be addressed before the manuscript can be accepted for publication.

1) My main concern is that when it comes to explaining the projected changes in O2, the authors did not explore all possible factors and focused nearly exclusively on the changes in water mixing ratios. While I believe that this likely plays a critical role in the transport and redistribution of O2 as suggested by the authors, other mechanisms that could also play a role were not considered. This includes for instance: 1) the changes (reduction) in biological export and respiration, 2) changes in winter stratification and convection (that could potentially be important in the deoxygenation signal that emerges in the northern Arabian Sea, alongside the effect of marginal seas warming).

(Response R1C1) We agree that the role of biological export and stratification changes should be addressed. We propose to add a subsection to our *Results* that will present the following points:

The mixing model experiments suggest that nearly all of the simulated oxygen changes in the SEC and northern Arabian Sea can be accounted for by shifts in water mass compositions driven by changes in advective ventilation pathways (Fig. GAC3). Oxygen changes captured by the mixing model generally agree within about 5 umol/kg of the total simulated oxygen change for most models (with the exception of the MPI models), and this result is robust to small changes in the definitions of source water

mass types (Fig. GAC3 presents a composite of 50 runs with perturbed source water properties; see new methods for OMP). The residual oxygen changes that the mixing model does not capture, while small, tend to be systematically biased positive for the western SEC and negative for the northern Arabian Sea. This suggests the influence of additional processes not accounted for in the mixing model. These processes include increases in subsurface stratification which can influence the vertical mixing between different water masses and changes in biological oxygen consumption after a water parcel leaves its source water region. We examine trends in stratification under SSP5-8.5 forcing (Figure GAC1). Strong stratification increases are generally confined to the upper 100 m of the water column. However, in both the Arabian Sea and SEC regions, the multi-model mean simulates significant stratification increases between 100-200 m. These stratification increases are simulated consistently across the ESM ensemble for both regions (*Individual model stratification trends in supplementary*). Across the ensemble, most of the oxygenation in the SEC region occurs below 200 m, where stratification does not show strong changes, suggesting that changes in vertical mixing likely play a relatively minor role in this region. In contrast, most of the deoxygenation in the northern Arabian Sea is collocated with a stratification increase between 100-200 m, suggesting that reduced vertical mixing of oxygenated surface waters downward contributes to the deoxygenation simulated in this region.

Increased subsurface stratification tends to limit the vertical mixing of oxygenated surface waters downward, but also limits the mixing of subsurface nutrients into the surface. This in turn limits primary productivity and the export of organic matter to the subsurface, setting a control on oxygen consumption rates at depth. We examine trends in the export of organic carbon at 100 m (available for 7 out of 8 ESMs; Figure GAC2). The export of organic carbon consistently declines under SSP5-8.5 forcing over the Indian Ocean in the multi-model mean, with local maximum declines in the western SEC and Arabian Sea regions. Declines in export are particularly strong in the Arabian Sea in 4 of 7 ESMs (MIROC-ES2L, MPI-ESM1-2-HR, MPI-ESM1-2-LR, UKESM1-0-LL) and in the western SEC in 2 of 7 models (MIROC-ES2L, UKESM1-0-LL; *Individual model export trends in supplementary*). The Arabian Sea and SEC experience both a significant decline in export rates and increased stratification at depth with opposing effects on oxygen changes, and thus are perturbed by processes not captured by the mixing model. The signs of residual oxygen changes not fit by the mixing model (Fig. GAC3) suggest that decline in export dominates over stratification increases in the SEC, and vice versa in the northern Arabian Sea. Though compensation between the effects of export and stratification changes may be key to allowing the mixing model to perform well in both regions. While the mixing model results plausibly show that the pattern of oxygen trends simulated by our ESM ensemble can be driven by changes in advective ventilation pathways, a quantitative decomposition of contributions from advective ventilation, biological sink and stratification changes is not possible with available data.

We also add related points in discussion section 4.4 addressing limitations of our approach, particularly in the Arabian Sea:

There are also aspects of the Arabian Sea which may limit the accuracy of the mixing model (Optimum Multiparameter; OMP) analysis employed in this study. Specifically, it may be difficult to distinguish the contribution of Persian Gulf Water compared to Arabian Sea surface water because both tend to be anomalously warm, saline and well oxygenated. Thus, it is possible that weakened vertical exchange with the surface is aliased as a decrease in Persian Gulf Water fraction. The presence of Northern Arabian Sea deoxygenation in MIROC-ES2L, an ESM which does not resolve the marginal seas and was thus excluded from the OMP analysis for the region, also suggests that stratification increases may play a first-order role in Northern Arabian Sea deoxygenation in the ESM ensemble. A previous study by Lachkar et al. (2021), which investigates deoxygenation in the Arabian Sea from 1982 to 2010 using a high-resolution regional ocean model, attributes about 75% of ventilation decreases over this period to vertical mixing changes and only 25% to advective changes.

2) My second concern is that most of the analysis is focused on the presentation of multi-model means (MMM). While this is useful to derive average patterns among models, it masks out the behavior of individual models and hides important differences that can be useful to document and understand. For instance, do models that represent complex biogeochemical feedbacks behave differently from those that don't when it comes to future O2 changes? Are there models where the oxygenation/deoxygenation patterns (or the drivers of those changes) are qualitatively different from the MMM despite having a decent representation of present-day O2 conditions in the Indian Ocean? And if so, why? It would be interesting to explore individual model responses in addition to the MMM to gain a broader understanding of the uncertainties around projections. The individual models can be shown in the main paper (for example for curve plots) or in the supplementary information (for maps or vertical sections) and discussed in the main paper.

(Response R1C2) This point refers to our General Response point 3 above. In the submitted manuscript, we used stippling over the MMM to highlight where individual models agree and disagree, and included maps and sections of individual models in Supplementary Information (Figures S1,S2, S6 and S7, submitted manuscript). Yet, we understand that adding information and discussion about individual models is useful to the reader. First, we propose to extend the discussion comparing the pattern of oxygen trends across ESMs in section 3.3, based on Figure S7, leveraging comparisons within ESM "families":

"All three features of dissolved oxygen trends are captured in all individual ESMs in the ensemble (Fig. S6,S7), leading to a robust pattern in the multi-model mean (more than 75% of models agree on sign of trend; Fig. 3). However, there are some notable differences in the amplitude and extent of simulated oxygen trend features across the ensemble. We can leverage comparisons between related models to illustrate the sensitivity of the oxygen response amplitude to the model representations (e.g. resolution and biogeochemical complexity). For example, GFDL-CM4 and GFDL-ESM4 have the same dynamical ocean model but differ in both oceanic and atmospheric resolution (GFDL-CM4 has higher resolution) and biogeochemical components (GFDL-ESM4 has more complex representation of biogeochemistry). Despite very similar historical representations of oxygen in the Indian Ocean, we find that GFDL-CM4 and GFDL-ESM4 simulate different strengths of oxygenation in the SEC (Feature 2) and deoxygenation in the Arabian Sea (Feature 3), with GFDL-CM4 projecting more pronounced features in both regions. Meanwhile, NorESM2-LM and NorESM2-MM, which differ only in the horizontal resolution of their atmospheric components, project similar trends for oxygenation in the SEC but differ for deoxygenation in the Arabian Sea, with weaker deoxygenation simulated in the higher resolution version (NorESM2-MM). The influence of model biogeochemical complexity is also demonstrated by CanESM5 and CanESM5-CanOE, both of which were removed from our multi-model mean due to the absence of suboxia but we use them here to illustrate the sensitivity to the model set up. CanESM5 and CanESM5-CanOE, which only differ in the complexity of their in biogeochemical model, project different strengths for both Features 2 and 3, with CanESM5-CanOE (more complex representation of biogeochemistry) projecting weaker oxygenation in the SEC and stronger deoxygenation in the Arabian Sea. These results show that the sign of oxygen change is robust across the ensemble, but that both resolution and complexity of biogeochemistry influence the amplitude of the change."

We also propose to add individual model curves to Figures 2 and 3 (OMZ volume and OMZ volume trends versus oxygen threshold; Figures GAC4,5,6). For Figure GAC5 in particular, we see that 7 of 8 ESMs simulate contraction for hypoxic thresholds and lower (~0-60umol/kg), and the model spread is coming primarily from a single model which does not simulate contraction (MIROC-ES2L).

3) Finally, OMZs are traditionally connected to hypoxia but some thresholds used in this study to characterize the OMZ (e.g. 150 mmol/m3) are much higher than the values typically used to define the OMZs of the Indian Ocean. As a result, regions that are far from the Arabian Sea and Bay of Bengal are also considered a part of the OMZ (e.g., the equatorial Indian Ocean and regions as south as 20S in the eastern Indian Ocean). Furthermore, some of the analyses (Figs 2, 3, 11) refer to the OMZ volume but show O2

concentrations as high as 225 mmol/m3. I suggest rephrasing the presentation to make it clearer that O2 changes in the entire upper ocean are analyzed and not just the OMZ.

(Response R1C3) We understand the reviewer's perspective and this is a point the coauthors discussed when writing the manuscript. The "OMZ volume" is a familiar metric to the Biogeosciences readership, which implies a volume of water below a defined o2 threshold. In contrast, a more general term such as "ocean volume" or "cumulative ocean volume" may be less intuitive. As mentioned in our General Response point 2, definitions of low-oxygen environments are highly dependent on the system of interest, e.g., many macroorganisms experience sublethal threats at oxygen levels of 150 umol/kg including most tunas found in the Indian Ocean (eg Dueri 2017; see lines 133-138 in submitted manuscript). In addition, we believe that the terms OMZ20, OMZ60, and OMZ150 used in the text guide the readers and avoid confusion as they clearly state the oxygen threshold. To address the reviewer's concern, we propose to further clarify in methods that the highest thresholds presented are not generally used to define the boundary of the OMZ at the end of section 2.2.1: "We note that oxygen thresholds above 150 umol/kg are not generally used to delimit low oxygen environments and OMZs, but we present them here to provide a holistic view of forced changes in oxygen distribution."

**Specific comments:**

Lines 92-101 (model selection): The figures in the Supp Info provide a visual evaluation with a horizontal map and a vertical section in the AS. A more quantitative approach that considers for instance the volumes of key O2 classes (e.g. below 20, or 60) would provide a more objective basis to retain or reject models. For instance, the UKESM model may not necessarily be better at simulating the OMZ (very thin and shallow) relative to the CNRM model (that was excluded). A related question is how sensitive the results are to this model selection. More generally, I think using multi-model means (MMM) can mask out systematic biases in models (for instance all models underestimate (overestimate) the intensity of the Arabian Sea (Bay of Bengal) OMZ and is not ideal to understand the sources of discrepancies between the multiple models and hence the uncertainties around future projections. This point needs to be at least discussed. One way to reduce an excessive reliance on MMM is to present the results from individual models whenever possible.

Thank you for the comment. We agree that a discussion of sensitivity to model selection would be useful. The spatial pattern of oxygen trends that we identify in the MMM (oxygenation/deoxygenation features) is not qualitatively sensitive to our model

selection, though models simulate a spread in the amplitude of these features (Figure R1AC1, to be added to SI). However, the analysis of OMZ volume and "contraction/expansion regimes" is obviously influenced by models with strong biases. For example, OMZ core volumes in the Arabian Sea cannot shrink in models that simulate no OMZ core in the first place. It is therefore critical to select models that represent certain historical OMZ features to examine their projections. We propose to take your solution to address this in a more objective way, and exclude models that do not simulate a suboxic (<10umol/kg) volume in the Arabian Sea (Table R1AC1).

Lines 99-101: It would be good to mention other sources of biases such as the representation of the Persian Gulf, the remineralization depth, the ill-represented mesoscale eddy activity, the misrepresented upwelling and productivity in these coarse resolution models.

These are important sources of biases in simulating oxygen in the Arabian Sea and Bay of Bengal, and we propose to add these points to section 3.1: "The overestimation of simulated oxygen values in the Arabian Sea (Fig GAC4b) is likely due primarily to excessive ventilation from marginal seas and southern source waters, though may also be influenced by deficiencies in parameterized oxygen consumption rates and eddy-mixing rates (eg Schmidt et al, 2021). In contrast, ESMs tend to simulate lower than observed oxygen values in the Bay of Bengal (Fig GAC4c), likely because of an overestimation of oxygen consumption rates and remineralization depths (Al Azhar et al., 2017)."

Lines 104-105: Maybe justify the use of a single member instead of the ensemble average (given the importance of forced vs. internal variability)?

The representation of OMZs, or more broadly volume as a function of oxygen, is so different across CMIP-class ESMs that model uncertainty tends to dwarf internal variability (eg Busecke et al. 2022). So, we chose to use only a single member for each model to minimize computational costs (by about an order of magnitude compared to using ensemble averages). Additionally, 7 of 14 models only have one member available with oxygen. We propose to modify this line to justify: "To limit computational costs of this study, and because we expect inter-model variability in dissolved oxygen to dominate over internal variability, only one member is used for each model."

Line 116: You mean the 85-year trend?

Yes, this should read 85-year (not 75-year). This will be corrected here and elsewhere. Thank you for catching this.

Lines 132-134: Since the OMZ is located below 100m, why include the surface (0-100) layer? I suggest excluding the surface layer in the analysis of O2 classes' volumes. Figure 2: Can you

redo this analysis after excluding the top 100m? Since the OMZ is located in the thermocline, it makes more sense to exclude the surface layer.

By definition, the oxygenated surface layer is excluded from the OMZ volume for sufficiently low oxygen thresholds, as demonstrated in Figure R1AC2. Thus, expliciting excluding the surface does not change the analysis.

Besides, this sort of analysis mixes up the OMZ in the Arabian Sea (AS) and in the Bay of Bengal (BoB). As most models overestimate O2 in the AS and underestimate it in the BoB, integrating the OMZ volumes across the entire IO can artificially hide those biases as these errors tend to compensate. It would be good to redo the same figure separately for the AS and the BoB.  Also Lines 232-233: This is an important difference (and I suspect it is worse in the BoB). It would be good to show these biases in a table while separating the AS and the BoB.

Yes this is a good point. We will add a figure with separate volumes in the AS and BoB (Fig GAC4,6), as well as a discussion in the main text in section 3.1 in order to clarify the interpretation of the model evaluation presented in Figure 2. We also propose to add a supplementary table (Table R1AC1) with separate volume values for the Arabian Sea and Bay of Bengal.

Line 155: Why is only the transport through the Timor passage considered? Why not the total ITF?

We compute transport along 114E along a section connecting Australia and Indonesia. While this section is dominated by the Timor passage, it actually captures transport through the Lombok and Ombai straits as well. For the coarse resolution models used here, this is a good approximation for the total ITF transport. The only other passage to consider in these models that is not captured at 114E is the Sunda strait, which tends to be represented as a single cell open to only 50 m depth, and does not contribute to the transport between 100-1000m. These points will be clarified in the text.

Lines 184-186: Why not take a mean over the last 20 years or so to represent the climatology for the end of the century period instead? justify.

The 'delta' method that we use here (perturbing a historical climatology by a forced linear trend) produces a climatological representation of the end-of-the-century, similar to taking a mean over the last 20 years. A benefit of the delta method is that the trend approximates forced changes at the very end of the time period. In contrast, a mean over the last 20 years may damp the amplitude of the forced signal (one misses the last decade of forced changes).

Line 204: Full stop before "Second".

Thank you.

Line 227: Can you show this analysis for individual models as well? not just the MMM? You can also show individual model based biases in a table for the key classes: O2<5, O2<20, O2<60, O2<150 mmol/m3.

Yes, we will add curves for individual models to Figure 2 (see Figure GAC4) and include Table R1AC1 in supplementary information.

Lines 294-295: Why not diagnosing trends in O2 consumption or remineralization? if the remineralization fluxes are not available, trends in export fluxes at 100 and 500m can be considered. Furthermore, the drivers of O2 interannual variability may differ from those of long-term O2 changes. Therefore, I am not fully convinced that the correlation really demonstrates the sources of the long-term oxygenation/deoxygenation. The biology can clearly play a role as models show a robust and important decline in the tropical Indian Ocean productivity by the end of the century. In any case, the two factors (ventilation and respiration) need to be equally explored. Lines 303-304: Same remark. Why is the role of biology not explored or discussed?

We propose to include an analysis of trends in export at 100m as part of an additional results section: *3.7 Contributions of biological sink and stratification changes to oxygen trends* (see Response R1C1).

We agree that the correlation of AOU:age may reflect short-term variability rather than a long term trend, and plan to remove those panels and associated text (Figure 6def, original manuscript).

Lines 311-314: How about the O2 content of these water masses? a change in the O2 content of these water masses can nullify those changes in the mixing ratios. This critical point needs to be explored. The ventilation depends both on the circulation/mixing as well as the O2 content of the water masses (e.g., pre-formed O2). The transport of O2 needs to be diagnosed.

This is a very good point. We performed the additional analysis of tracking the O2 transport of these water masses over time, and we find that the basin-scale oxygen supply via the Southern Pathway and ITF declines despite the shifting water mass ratio (Figure R1AC3). Thus, the timeseries of water mass transport ratios in 7b does not demonstrate (as originally argued) that oxygen supply increases on a basin-wide scale. We plan to remove Figure 7b from the main text as it may be misleading. Rather, it is only changes in mixing ratios at more local scales (as discerned by the OMP analysis) that are sufficient to overcome O2 changes in each source water mass and lead to increased oxygen supply locally.

Lines 329-331: This assumes that these O2 concentrations remain constant under future climate change, which I am not sure is true.

Yes, we agree that the proper calculation of mixing-driven O2 changes should account for changes in source water O2 concentrations between the historical and future climatologies. We propose to add Figure GAC3 to the results sections 3.5 and 3.6. We will add a description of the associated method in section 2.2.4 (See methods at end of General Response). Calculations of mixing-driven O2 changes yield a wide range of values across models in the ensemble, ranging between about 10 to 40 umol/kg change between the historical and end-of-century climatologies in the South Equatorial Current. Similarly, mixing-driven O2 changes range from about -100 to -25 in the northern Arabian Sea. In both regions, mixing-driven O2 changes agree well with total O2 changes, demonstrating that the mixing model can plausibly account for oxygen trends via shifting water mass compositions.

We will remove the back-of-the-envelope calculation in lines 329-331.

Lines 374-379: Need to cite the recent work by Vallivattathillam et al. (2023, Frontiers in Mar. Sc.) who also show a robust shrinking of the OMZ in the Arabian Sea based on the analysis of a set of CMIP5 models as well as downscaled regional model simulations. Line 430: I suggest citing two recent studies of the Arabian Sea OMZ that discussed some sources of uncertainties in future model OMZ projections: Vallivattathillam et al. (2023, Frontiers in Mar. Sci.) and Lachkar et al., (2023, Frontiers in Mar. Sci.). Vallivattathillam et al. (2023) explored the sensitivity of future O2 projections to model representation of present-day conditions and highlighted the importance of correcting biases in global models representation of present-day OMZs. This appears to considerably reduce the discrepancies among models and makes the OMZ shrinking signal emerge more robustly. Lachkar et al (2023) discussed a few factors (e.g., timescales of the different mechanisms) that may explain the difference between recent trends dominated by deoxygenation in the AS and the projected future changes dominated by oxygenation.

Yes, we agree that these works are very relevant to our study. We propose to expand our discussion of Arabian Sea projections (lines 442-450 in submitted manuscript) as follows:

"Deoxygenation in the northern Arabian Sea has been detected in observations (reviewed by Lachkar et al., 2023). The rapid heating of marginal seas compared to the open ocean and the subsequent increase in stratification and buoyancy of the marginal sea outflows, particularly for the Persian Gulf is also supported by observations (Al-Yamani et al., 2017; Naqvi, 2021). The projected changes in outflow ventilation are consistent with the findings of ocean model simulation studies, which showed that the warming and shoaling of the Persian Gulf was a major driver of the deoxygenation in

the northern Arabian Sea (Lachkar et al., 2019, 2021). Yet, ESMs tend to overestimate the oxygen transport by these marginal sea outflows to the Arabian Sea (e.g. Schmidt et al., 2021). This partially explains the systematic high oxygen bias in the Arabian Sea in the ensemble of eight CMIP6 ESMs used in this study (Fig. 1), the absence of a suboxia in the five ESMs that were excluded from the ensemble (Table R1AC1), and could influence the magnitude of the projected deoxygenation associated with the vertical displacement of these outflows. In fact, a study by Vallivattathillam et al. (2023), which applies downscaling methods to CMIP5 projections of Arabian Sea oxygen under the high emission RCP8.5 forcing, finds that deoxygenation in the region simulated in CMIP5 models is not preserved after a bias-corrected downscaling method is applied (sampling between 200 and 700 m). This suggests that ESM simulated mean state biases in the Arabian Sea can have a profound impact on projected changes.

There are also aspects of the Arabian Sea which may limit the accuracy of the mixing model (Optimum Multiparameter; OMP) analysis employed in this study. Specifically, it may be difficult to distinguish the contribution of Persian Gulf Water compared to Arabian Sea surface water because both tend to be anomalously warm, saline and well oxygenated. Thus, it is possible that weakened vertical exchange with the surface is aliased as a decrease in Persian Gulf Water fraction. The presence of Northern Arabian Sea deoxygenation in MIROC-ES2L, an ESM which does not resolve the marginal seas and was thus excluded from the OMP analysis for the region, also suggests that stratification increases may play a first-order role in Northern Arabian Sea deoxygenation in the ESM ensemble. A previous study by Lachkar et al. (2021), which investigates deoxygenation in the Arabian Sea from 1982 to 2010 using a high-resolution regional ocean model, attributes about 75% of ventilation decreases over this period to vertical mixing changes and only 25% to advective changes. In contrast, Vallivattathillam et al. (2023) demonstrate that remote physical forcing (including marginal seas) drives deoxygenation below 200 m in the northern Arabian Sea while local forcing (including stratification from local warming) does not in downscaled projections under RCP8.5 forcing.

Further work, including for example a full oxygen budget analysis, would be required to better disambiguate the effects of marginal sea shoaling and increased stratification in the projections of Arabian Sea oxygen for the ESMs used in this study; however, data for such a study are not currently available. Based on these considerations, the relative effects of marginal sea shoaling and increased stratification in ESM projections of the northern Arabian Sea remain largely uncertain. Ultimately though, the present study supports the conclusions of the recent review by Lachkar et al. (2023): the Arabian Sea OMZ will be reshaped by changes in stratification and the Persian Gulf outflow driving deoxygenation in the northern subsurface layer, while increased ventilation from the south drives oxygenation of the deeper layers."

Fig S1: Maybe show the difference model-minus-obs to better visualize individual model biases.

Yes, this is helpful for visualizing individual model biases (see Figure R1AC4). We will include it in supplementary information to accompany Figure S2 (original manuscript).

Fig S5: You mean Figure 1 in the main paper?

Yes, thank you for catching this.

Fig S6: Are these trends vertically averaged as stated in the caption, or are they trends in the vertically averaged O2 concentrations?

Figure S6 shows the trends vertically averaged, as stated in the caption. We first calculate a 3D field of oxygen trends for each model regridded to a rectilinear 1x1 degree grid, then take a vertical average. Performing operations in this order allows for a direct comparison between maps as in Figure S6 and sections as in Figure S7.

The remaining specific comments of Reviewer 1 are all referring to main concern #2 in the general response (definition of OMZ). Please refer to our General Response point 2, and to Response R1C3.

Lines 9-10: Is O2 at 180 considered part of the OMZ?

Lines 132-138: It is not clear which OMZ definition this paper uses. OMZs are traditionally connected to hypoxia, but some thresholds used here (150 mmol/m3) are way above hypoxia, and result in nearly ⅔ of the Indian Ocean being filled with OMZ waters (down to 20S). This is very unusual as the Indian Ocean OMZs are traditionally restricted to the Arabian Sea and the Bay of Bengal in the northern Indian Ocean. It is true that some previous studies suggested that some commercial fish species can be stressed at O2 levels around 150 or below. However, all these works are based on studies of the Atlantic and Pacific Oceans and hence may not necessarily be relevant to the Indian Ocean species. The use of this threshold in the Indian Ocean needs to be justified and its implications discussed.

Figure 3: This should not be titled "OMZ volume changes" since O2 thresholds as high as 200 and above are used. You are sampling the entire upper ocean (0-1000m) not just the OMZ.Indeed, a significant portion of the analysis is dedicated to relatively well oxygenated waters, well above hypoxia. Therefore, I suggest rephrasing the presentation to make it clearer that O2 changes in the entire upper ocean are analyzed and not just the OMZ.

Lines 235-236: Given the very high threshold (180 mmol/m3), can one still refer to this as a part of the OMZ? This is likely in the surface layer and I suspect is essentially driven by solubility. Please rephrase.

Lines 239-245:Maybe refer to the different classes of O2 as O2<20, O2<60, O2<150,...instead of OMZ20, OMZ60, and OMZ150.

Figure 11:Again, can we talk about OMZ when O2 concentration is close to saturation (for O2 close to 200 mmol/m3)? This essentially shows the cumulative changes in O2 frequency distribution.

[Figure]

Figure R1AC1: Recreation of dissolved oxygen trends in Figure 4 using all 14 available ESMs. Multi-model mean (MMM) dissolved oxygen trends under SSP5-8.5 scenario forcing (1915-2100). Dissolved oxygen trends (a) bewtween 100 and 1000m, (b) at 65◦E and (c) at 90◦E. Dashed gray lines mark (a) 65◦E and 90◦E, and (b,c) 100 and 1000m. (b,c) Solid black contours represent 20, 60 and 150μmol/kg oxygen.

[Figure]

Figure R1AC2: Upper ocean OMZ volume as a function of oxygen threshold including and excluding upper 100 m.

[Figure]

Figure R1AC3: Interannual transport anomalies (referenced to 1950-2015 mean) for inflowing components of Southern Pathway Waters (blue) and Indonesian Throughflow Water (orange) between 100 and 1000 m. (b) Ratio of total inflowing transport of Southern Pathway andIndonesian Throughflow Waters. (c) Mean oxygen concentration of combined Southern Pathway and Indonesian Throughflow waters.

Solid curves represent multi-model mean (MMM) and shading represents one
standard deviation of model spread. Transport data is available for 5 ESMs.

[Figure]

Figure R1AC4: Oxygen bias relative to World Ocean Atlas at 65E. ESM names in red
are not included in the selected ensemble.

| | model | 10 (AS) | 20 (AS) | 20 (BB) | 60 | 150 |
|---|---|---|---|---|---|---|
| 0 | WOA | 0.2 | 2.1 | 0.6 | 11.4 | 62.9 |
| 1 | GFDL-CM4 | 0.4 | 0.7 | 1.5 | 7.1 | 43.2 |
| 2 | GFDL-ESM4 | 0.4 | 0.7 | 1.6 | 5.9 | 47.1 |
| 3 | MIROC-ES2L | 0.5 | 0.8 | 1.1 | 5.9 | 39.4 |
| 4 | MPI-ESM1-2-HR | 2.0 | 3.0 | 3.8 | 18.3 | 73.5 |
| 5 | MPI-ESM1-2-LR | 2.7 | 3.9 | 3.0 | 16.7 | 54.2 |
| 6 | NorESM2-LM | 1.7 | 2.4 | 3.2 | 13.7 | 42.0 |
| 7 | NorESM2-MM | 2.5 | 3.3 | 3.2 | 14.8 | 43.6 |
| 8 | UKESM1-0-LL | 0.3 | 0.6 | 0.7 | 4.4 | 18.3 |
| 9 | ACCESS-ESM1-5 | 0.0 | 0.0 | 0.4 | 2.8 | 40.5 |
| 10 | CanESM5 | 0.0 | 0.0 | 0.4 | 1.6 | 19.9 |
| 11 | CanESM5-CanOE | 0.0 | 0.0 | 0.9 | 4.1 | 35.8 |
| 12 | CNRM-ESM2-1 | 0.0 | 0.5 | 1.7 | 13.3 | 114.9 |
| 13 | IPSL-CM6A-LR | 0.0 | 0.0 | 0.1 | 0.7 | 105.7 |
| 14 | MRI-ESM2-0 | 0.0 | 0.0 | 1.3 | 3.9 | 60.5 |

Table R1AC1: OMZ volumes (in $10^{15}m^3$) for Word Ocean Atlas (WOA) and CMIP6 ESMs at thresholds of 10, 20, 60 150 umol/kg. Volume for thresholds of 10 and 20 umol/kg are separated as Arabian Sea (AS) and Bay of Bengal (BoB). Southern extent of AS and BoB domains are taken at 5N.

---

## Author Comment (AC2)

**General Response**

We thank the three reviewers for their constructive and insightful comments. The main concerns among the reviewers fall into 3 categories. We give a brief overview of how we propose to address them and include the proposed new figures here in the general response, and then provide a detailed response below.

1. **Role of changes in stratification and productivity in dissolved oxygen projections.** The reviewers ask for an expanded investigation of potential changes in upper ocean stratification and the biological sink on oxygen changes. Specifically, increased stratification could contribute to deoxygenation in the northern Arabian Sea, and decreased productivity and respiration rates could contribute to oxygenation in the South Equatorial Current (SEC). We propose to add a subsection to our *Results* in which we examine trends in upper ocean stratification (Figure GAC1) and the export of organic carbon at 100 m (Figure GAC2). Both the northern Arabian Sea and SEC regions are projected to experience increases in stratification between 100 - 200 m, as well as decreases in export at 100 m, with opposing effects on oxygen. The spatial correlation of these two trend fields is expected, given the relationship between stratification and nutrient supply via vertical mixing. This suggests that the effects of stratification and export trends, at least, partially compensate each other in both of these regions. We propose to discuss this compensation and how it is consistent with the results of our sections 3.5 and 3.6 (using Optimum Multiparameter (OMP) analysis) which suggests that changes in circulation and water mass fractions related to lateral advective pathways can explain nearly all simulated oxygen changes (direct calculations of mixing-driven oxygen changes will be added in the revised manuscript; Figure GAC3). We have also performed additional sensitivity tests with the OMP method (see new Figure GAC3 and new Method details at end of general response) and propose to add a discussion of the robustness and limitation of the OMP method, and the implications for our results (to be added to *Discussion* section 4.4).

2. **Clarification of approach.** One reviewer is concerned with the definition of Oxygen Minimum Zone (OMZ) used in the manuscript which is a continuum covering a broad range of oxygen values extending beyond the commonly used hypoxic thresholds (e.g., 60 umol/kg). We propose to clarify in the *Methods* section 2.2.1 and throughout the text the rationale for using this broad definition: (1) The relevant thresholds for 'low oxygen' environments can vary significantly depending on the application, such as (in order of increasing threshold) denitrification, mortality of marine organisms, and sublethal stresses on marine organisms. Specifically, thresholds of ~150 umol/kg (and even as high as 200 umol/kg) are commonly used as habitat boundaries for large commercial fish species (eg Vaquer-Sunyer and Duarte, 2008); (2) Using a broad continuum of OMZ thresholds enables us to interpret changes in oxygen volume (including the hypoxic volume) within the context of larger-scale forced changes (contraction, redistribution, expansion).

3. **Robustness of findings across individual Earth System Models (ESM)**. The reviewers note that much of the analysis presented in the manuscript is based on the presentation of Multi-Model Means (MMM) of the ESM ensemble, which may mask

compensating biases and different dynamical responses across models and may be sensitive to model selection. We propose to a) expand the discussion of the maps/sections of individual ESMs already included in the Supplementary Information (SI) of the original manuscript, b) add individual ESMs when presenting OMZ volume and OMZ volume trends in the main text (new Figures GAC4, GAC5, GAC6), and c) compare the effects of water mass fraction changes (from OMP analysis) across individual models (Figure GAC3). Overall, we find that most models exhibit the same regional pattern of oxygen trends and regimes of OMZ volume trends as the MMM, with varying amplitudes of features across models. This will also be discussed in Discussion section 4.4.

We are confident that these revisions will address the concerns of the reviewers. Below, we provide specific details and figures for these revisions, and we address additional reviewer comments.

[Figure]

Figure GAC (General Author Comment) 1: Multi-model mean (MMM) stratification trends under SSP5-8.5 scenario forcing (1915-2100). Stratification trends (a) between 100 and 200m, and (b) at 65E. (c) dissolved oxygen trends at 65E. (b,c) Solid black contours represent 20, 60 and 150 μmol/kg oxygen. Dashed black contours highlight salinity signature of Subtropical Underwater (STUW). Results are stippled where less than 75% (6/8) of models agree on sign of trend.

[Figure]

Figure GAC2: Multi-model mean (MMM) export of organic carbon at 100 m trends under SSP5-8.5 scenario forcing (1915-2100). Results are stippled where less than 75% (6/7) of models agree on sign of trend.

[Figure]

Figure GAC3: Changes in dissolved oxygen accounted for by mixing model analysis. Total oxygen changes (simulated by ESMs) versus oxygen changes attributed to shifts in water mass composition in the OMP mixing mode for the (a) western South Equatorial Current (SEC) and (b) northern Arabian Sea. Results in the western SEC averaged between 40-80E, 5-15S. Results in the northern Arabian Sea averaged between 62-70E, 20-26N. Solid markers are the mean of 50 runs (light markers) of the OMP analysis with perturbed source water properties (See end of general response for details on new methods). Black dashed line marks 1-to-1 relationship, gray dashed lines mark 5 umol/kg deviations from 1-to-1.

**New method to be added to Optimum Multiparameter (OMP, section 2.2.4) analysis for sensitivity analysis and used to produce Figure GAC3 above**:

"From the mixing model results, we quantify the change in oxygen accounted for by shifts in water mass composition for key regions: $\Delta O_{mixing} = \sum_i (f_{i,1} O_{i,1} - f_{i,0} O_{i,0})$, where $f_i$ and $O_i$ are the fraction and oxygen concentration of source water type $i$, for future (1) and historical (0) states. Oxygen is allowed to be remineralized, so $\Delta O_{mixing}$ need not approximate the total simulated change in oxygen. To test the sensitivity of the mixing model results to small changes in the definition of source water types and potential density layer, we average 50 realizations of the experiment applying random perturbations to source water locations and density values. Source water locations are perturbed by up to 5 degrees in latitude and longitude for the South Equatorial Current region and 2 degrees for the Arabian Sea. For both locations, the value of the potential density layer is perturbed by up to 0.1."

We note regarding the OMP analysis that we lowered the weight on oxygen in the optimization compared to the submitted manuscript (lowered from 20% the weight of temperature and salinity to 2%) to more appropriately account for the range of oxygen values in the analysis compared to the range and temperature and salinities (see Appendix A of submitted manuscript for technical details of OMP).

[Figure]

Figure GAC4: (**To replace Figure 2 in main text**) OMZ volume taken between 0-1000 m in the (a) Indian Ocean and sub-basins (b) Arabian Sea and (c) Bay of Bengal for the multi-model mean averaged over 1950-2015 (MMM; black) and observed climatology (WOA18; red). Shading represents one standard deviation of the model spread. Individual ESMs shown in colored curves. Southern boundary of Arabian Sea and Bay of Bengal are at 5N.

[Figure]

Figure GAC5: (**To replace Figure 3 in main text**) Multi-model mean thermocline OMZ volume changes (between 0 - 1000 m) under SSP5-8.5 scenario forcing (2015-2100). (a) $V_{o2}^{1000}$ trends as a function of oxygen threshold for the Indian Ocean. The 20, 60 and 150 umol/kg thresholds bounding OMZ20, OMZ60 and OMZ150 are indicated with grey dashed lines. (b) Time series of $V_{o2}^{1000}$ anomaly from 1950-2100 (anomaly referenced to 1950-2015 mean) for OMZ60 (dark blue) and OMZ150 (teal). Shading represents one standard deviation of model spread.

[Figure]

Figure GAC6: (**To be added to supplementary information**) Multi-model mean thermocline OMZ volume changes (between 0 - 1000 m) under SSP5-8.5 scenario forcing (2015-2100). $V_{o2}^{1000}$ trends as a function of oxygen threshold for the (a) Arabian Sea and (b) Bay of Bengal with OMZ20 volumes indicated for each basin. (c) Time series of $V_{o2}^{1000}$ anomaly from 1950-2100 for OMZ20 volumes in the Arabian Sea and Bay of Bengal. Shading represents one standard deviation of model spread.

**Reviewer 2:**

Review of Ditkovsky et al.

Anand Gnanadesikan

This paper considers drivers of changes in hypoxia in the Indian Ocean, a critical region for artisanal fisheries and one whose behavior under global warming has not been well characterized. The paper finds three regimes of oxygen change, which correspond to three different driving mechanisms for that change. The authors kindly credit me with distinguishing "single-pipe" from a "mixing network" models and describe how this can be used to distinguish the changes in the northern, central, and southern Indian Oceans and how it needs to be updated to do so. Frankly, I think their mapping onto the different regimes is clearer than what I wrote about- this is an elegantly written paper. The argument that there are three separate regimes is generally well made.

I have three comments, that in some ways parallel those made by the other reviewer. Normally, I would give this a "major" revision since they will require some new analysis, not merely clarification of existing analysis. But I think the basic analysis is sound and don't want to suggest "reconsideration" of the paper, so I'll call this minor.

1. The first comment regards the role of the overflows. There is a lot more to overflows than the volume that they deliver to the ocean. It's been a problem for a long time in models to get the depths of injection of overflow water correct. If I look at the salinity along 70E (Figure R1), it's clear that there's a signal from high-salinity shelf waters that penetrates for hundreds of m. It is unlikely that models, which generally have problems with numerical entrainment, correctly capture either this process or its sensitivity to changes in climate. This is even true for relatively high-resolution models (see Seddigh-Marvasti et al., 2015 for some discussion of this). It would be good to at least evaluate how much of a problem this is, rather than simply accepting the results of the MMM (cross-sections of the watermass fractions might be useful to look at this and discern whether there are any systematic errors here). This wouldn't need to be an extra figure in the main text but would make a good one in the Supplemental material and would be useful for evaluating whether there are any systematic errors here.

      Indeed, the Arabian Sea is a region of strong salinity and oxygen bias in ESMs in large part due to overflows (eg Schmidt et al., 2021). A comparison of salinity sections at 65E (Figure R2AC1 below) suggests that there is significant inter-model variability in the strength of the overflow signals, with most models simulating too-saline subsurface waters compared to WOA. However, our model selection process, based on an oxygen criterion does mitigate the salinity bias in the Arabian Sea, as well as the oxygen bias. We propose to extend our explanation of model selection in section 2.1 as follows:

"Out of the 14 CMIP6 ESMs that provided monthly dissolved oxygen data for the pre-industrial control, historical, and SSP5-8.5 experiments, we exclude 6 models that simulate virtually no suboxic (<10 umol/kg) volume in the Arabian Sea (ACCESS-ESM1-5, CanESM5, CanESM5-CanOE, CNRM-ESM2-1, IPSL-CM6A-LR; Table R1AC1). We keep the 8 remaining ESMs (GFDL-CM4, GFDL-ESM4, MIROC-ES2L, MPI-ESM1-2-HR, MPI-ESM1-2-LR, NorESM2-LM, NorESM2-MM, UKESM1-0-LL). All six ESMs excluded from the multi-model mean exhibit above average salinity biases in the Arabian Sea from outflows (Fig. R2AC1) and four of six models exhibit Red Sea outflow rates over twice the observed rate (Fig. S6 *from original submission – Red sea outflow rates*}). Thus, the representation of marginal sea outflows may be improved significantly in our ensemble by excluding these ESMs."

While salinity biases in our ensemble are improved by model selection, they are certainly not eliminated. We note that there are compensating salinity biases between models in our selected ensemble that may mask issues when examining a multi-model mean, namely the fresh MIROC-ES2L (which does not have overflows) and the deep salinity biases of the NorESM2-LM and NorESM2-MM. A discussion of these biases will be included in section 3.1 on model evaluation.

Consideration of these biases does not change our analysis, but it will factor into the discussion section 4.4 on interpreting our results for the Arabian Sea. The extended revision of this section is attached in response to your third comment.

2. The neglect of changes in productivity is understandable, but I note that it was also picked up by the other reviewer. One way of addressing this is to look at how much of the change in the oxygen can be accounted for by changes in the O2:age slope and how much can be accounted for by changes in the age itself (i.e. to the extent that $AOU=JO_2*age \rightarrow \Delta AOU=JO_2*\Delta age + age*\Delta JO_2$) Showing not just the correlation coefficients but the regression coefficients might help with this.

We propose to address changes in productivity by including an analysis of trends in export of organic carbon at 100 m (see General Response 1). ΔJO2 may be a difficult quantity to interpret, as it is an average rate over the lifetime of a water parcel, and thus will be strongly influenced by remote changes outside of the Indian Ocean. The relationship between AOU and JO2 can also be distorted by mixing giving spurious results (e.g. Guo et al., GRL 2023), as well as issues with ideal age in CMIP6 ESMs (requirement of long spin-ups). Instead, we propose to add a subsection to our *Results*

that will present the following points regarding changes in the biological sink and stratification:

The mixing model experiments suggest that nearly all of the simulated oxygen changes in the SEC and northern Arabian Sea can be accounted for by shifts in water mass compositions driven by changes in advective ventilation pathways (Fig. GAC3). Oxygen changes captured by the mixing model generally agree within about 5 umol/kg of the total simulated oxygen change for most models (with the exception of the MPI models), and this result is robust to small changes in the definitions of source water mass types (Fig. GAC3 presents a composite of 50 runs with perturbed source water properties; see new methods for OMP). The residual oxygen changes that the mixing model does not capture, while small, tend to be systematically biased positive for the western SEC and negative for the northern Arabian Sea. This suggests the influence of additional processes not accounted for in the mixing model. These processes include increases in subsurface stratification which can influence the vertical mixing between different water masses and changes in biological oxygen consumption after a water parcel leaves its source water region. We examine trends in stratification under SSP5-8.5 forcing (Figure GAC1). Strong stratification increases are generally confined to the upper 100 m of the water column. However, in both the Arabian Sea and SEC regions, the multi-model mean simulates significant stratification increases between 100-200 m. These stratification increases are simulated consistently across the ESM ensemble for both regions (*Individual model stratification trends in supplementary*). Across the ensemble, most of the oxygenation in the SEC region occurs below 200 m, where stratification does not show strong changes, suggesting that changes in vertical mixing likely play a relatively minor role in this region. In contrast, most of the deoxygenation in the northern Arabian Sea is collocated with a stratification increase between 100-200 m, suggesting that reduced vertical mixing of oxygenated surface waters downward contributes to the deoxygenation simulated in this region.

Increased subsurface stratification tends to limit the vertical mixing of oxygenated surface waters downward, but also limits the mixing of subsurface nutrients into the surface. This in turn limits primary productivity and the export of organic matter to the subsurface, setting a control on oxygen consumption rates at depth. We examine trends in the export of organic carbon at 100 m (available for 7 out of 8 ESMs; Figure GAC2). The export of organic carbon consistently declines under SSP5-8.5 forcing over the Indian Ocean in the multi-model mean, with local maximum declines in the western SEC and Arabian Sea regions. Declines in export are particularly strong in the Arabian Sea in 4 of 7 ESMs (MIROC-ES2L, MPI-ESM1-2-HR, MPI-ESM1-2-LR, UKESM1-0-LL) and in the western SEC in 2 of 7 models (MIROC-ES2L, UKESM1-0-LL; *Individual model export trends in supplementary*). The Arabian Sea and SEC experience both a significant decline in export rates and increased stratification at depth with opposing

effects on oxygen changes, and thus are perturbed by processes not captured by the mixing model. The residual (non source water-driven) oxygen changes from the mixing model suggest that decline in export dominates over stratification increases in the SEC, and vice versa in the northern Arabian Sea. Though compensation between the effects of export and stratification changes may be key to allowing the mixing model to perform well in both regions. While the mixing model results plausibly show that the pattern of oxygen trends simulated by our ESM ensemble can be driven by changes in advective ventilation pathways, a quantitative decomposition of contributions from advective ventilation, biological sink and stratification changes is not possible with available data.

3. Finally, I wanted to see whether this picture seemed to work in my own suite of coarse models reported in Bahl et al., 2019. These models don't have a Red Sea or Persian Gulf, and also fail to generate an oxygen minimum zone in the Northern Arabian Sea (this, incidentally, supports the point of this paper and others that resolving the impacts of such water is important).I show results for two cases with low and high lateral mixing in the figure below. Interestingly, it does seem that the same 3 regimes show up.

Since I have a remineralized phosphate tracer in this model I can also directly attribute the changes to changes in accumulated phosphate, and again, this dominates the pattern at low mixing, though somewhat less at high mixing. Ultimately I think this highlights an interesting question of whether the real world Indonesian throughflow acts as a barrier to or enhancer of tracer mixing. It also raises the question of how much of the intermodel variability is due to how this subgridscale mixing is handled. That the basic framework seems to work in this model suite as well is encouraging and supports the publication of the manuscript. Note,however, that despite not having marginal seas, we still get a drop in the North. This seems to be driven by a shallowing of mixed layers (echoing something noted by the other reviewer), which reach over 100m on average in the winter in the Northern Arabian Sea in this model, but shallow substantially under global warming. This may be difficult to capture with the watermass analysis alone, as it is not clear (at least to me) that the MMM will necessarily capture the differences between overflow and surface watermasses in this region. It would be worth examining changes in mixed layer depth to see to what extent this plays a role in the more realistic models.

Thank you for providing these examples. It is interesting to see the same pattern of oxygen trends emerge, and the deoxygenation simulated in the Arabian Sea without overflows in your suite of models (this is also the case in MIROC-ES2L). We investigated forced changes in winter mixed layer depths (MLD)  in the Arabian Sea. We found that while the winter MLD shoals by up to 30 m in some models, the MLD was still too shallow (sitting around 100 m) to explain the volumes of deoxygenation. So

instead, we examined 3D fields of stratification trends more generally, and find that much of the deoxygenation is collocated with increased stratification in most models (see new results section above). We propose to interpret the relative influence of stratification changes versus changes in outflow ventilation – as well as other challenges in interpreting projections for the Arabian Sea –  as an expanded discussion in section 4.4:

"Deoxygenation in the northern Arabian Sea has been detected in observations (reviewed by Lachkar et al., 2023). The rapid heating of marginal seas compared to the open ocean and the subsequent increase in stratification and buoyancy of the marginal sea outflows, particularly for the Persian Gulf is also supported by observations (Al-Yamani et al., 2017; Naqvi, 2021). The projected changes in outflow ventilation are consistent with the findings of ocean model simulation studies, which showed that the warming and shoaling of the Persian Gulf was a major driver of the deoxygenation in the northern Arabian Sea (Lachkar et al., 2019, 2021). Yet, ESMs tend to overestimate the oxygen transport by these marginal sea outflows to the Arabian Sea (e.g. Schmidt et al., 2021). This partially explains the systematic high oxygen bias in the Arabian Sea in the ensemble of eight CMIP6 ESMs used in this study (Fig. 1), the absence of a suboxia in the five ESMs that were excluded from the ensemble (Table R1AC1), and could influence the magnitude of the projected deoxygenation associated with the vertical displacement of these outflows. In fact, a study by Vallivattathillam et al. (2023), which applies downscaling methods to CMIP5 projections of Arabian Sea oxygen under RCP8.5 forcing, finds that deoxygenation in the region simulated in CMIP5 models is not preserved after a bias-corrected downscaling method is applied (sampling between 200 and 700 m). This suggests that ESM simulated mean state biases in the Arabian Sea can have a profound impact on projected changes.

There are also aspects of the Arabian Sea which may limit the accuracy of the mixing model (Optimum Multiparameter; OMP) analysis employed in this study. Specifically, it may be difficult to distinguish the contribution of Persian Gulf Water compared to Arabian Sea surface water because both tend to be anomalously warm, saline and well oxygenated. Thus, it is possible that weakened vertical exchange with the surface is aliased as a decrease in Persian Gulf Water fraction. The presence of Northern Arabian Sea deoxygenation in MIROC-ES2L, an ESM which does not resolve the marginal seas and was thus excluded from the OMP analysis for the region, also suggests that stratification increases may play a first-order role in Northern Arabian Sea deoxygenation in the ESM ensemble. A previous study by Lachkar et al. (2021), which investigates deoxygenation in the Arabian Sea from 1982 to 2010 using a high-resolution regional ocean model, attributes about 75% of ventilation decreases over this period to vertical mixing changes and only 25% to advective changes. In contrast, Vallivattathillam et al. (2023) demonstrate that remote physical forcing

(including marginal seas) drives deoxygenation below 200 m in the northern Arabian Sea while local forcing (including stratification from local warming) does not in downscaled projections under RCP8.5 forcing.

Further work, including for example a full oxygen budget analysis, would be required to better disambiguate the effects of marginal sea shoaling and increased stratification in the projections of Arabian Sea oxygen for the ESMs used in this study; however, data for such a study are not currently available. Based on these considerations, the relative effects of marginal sea shoaling and increased stratification in ESM projections of the northern Arabian Sea remain largely uncertain. Ultimately though, the present study supports the conclusions of the recent review by Lachkar et al. (2023): the Arabian Sea OMZ will be reshaped by changes in stratification and the Persian Gulf outflow driving deoxygenation in the northern subsurface layer, while increased ventilation from the south drives oxygenation of the deeper layers."

[Figure]

Figure R2AC1: Section of historical mean salinity in the Arabian Sea at 65E in World Ocean Atlas (WOA) and CMIP6 models. Historical mean in CMIP6 models taken from 1950-2015.

---

## Author Comment (AC3)

**General Response**

We thank the three reviewers for their constructive and insightful comments. The main concerns among the reviewers fall into 3 categories. We give a brief overview of how we propose to address them and include the proposed new figures here in the general response, and then provide a detailed response below.

1. **Role of changes in stratification and productivity in dissolved oxygen projections.** The reviewers ask for an expanded investigation of potential changes in upper ocean stratification and the biological sink on oxygen changes. Specifically, increased stratification could contribute to deoxygenation in the northern Arabian Sea, and decreased productivity and respiration rates could contribute to oxygenation in the South Equatorial Current (SEC). We propose to add a subsection to our *Results* in which we examine trends in upper ocean stratification (Figure GAC1) and the export of organic carbon at 100 m (Figure GAC2). Both the northern Arabian Sea and SEC regions are projected to experience increases in stratification between 100 - 200 m, as well as decreases in export at 100 m, with opposing effects on oxygen. The spatial correlation of these two trend fields is expected, given the relationship between stratification and nutrient supply via vertical mixing. This suggests that the effects of stratification and export trends, at least, partially compensate each other in both of these regions. We propose to discuss this compensation and how it is consistent with the results of our sections 3.5 and 3.6 (using Optimum Multiparameter (OMP) analysis) which suggests that changes in circulation and water mass fractions related to lateral advective pathways can explain nearly all simulated oxygen changes (direct calculations of mixing-driven oxygen changes will be added in the revised manuscript; Figure GAC3). We have also performed additional sensitivity tests with the OMP method (see new Figure GAC3 and new Method details at end of general response) and propose to add a discussion of the robustness and limitation of the OMP method, and the implications for our results (to be added to *Discussion* section 4.4).

2. **Clarification of approach.** One reviewer is concerned with the definition of Oxygen Minimum Zone (OMZ) used in the manuscript which is a continuum covering a broad range of oxygen values extending beyond the commonly used hypoxic thresholds (e.g., 60 umol/kg). We propose to clarify in the *Methods* section 2.2.1 and throughout the text the rationale for using this broad definition: (1) The relevant thresholds for 'low oxygen' environments can vary significantly depending on the application, such as (in order of increasing threshold) denitrification, mortality of marine organisms, and sublethal stresses on marine organisms. Specifically, thresholds of ~150 umol/kg (and even as high as 200 umol/kg) are commonly used as habitat boundaries for large commercial fish species (eg Vaquer-Sunyer and Duarte, 2008); (2) Using a broad continuum of OMZ thresholds enables us to interpret changes in oxygen volume (including the hypoxic volume) within the context of larger-scale forced changes (contraction, redistribution, expansion).

3. **Robustness of findings across individual Earth System Models (ESM)**. The reviewers note that much of the analysis presented in the manuscript is based on the presentation of Multi-Model Means (MMM) of the ESM ensemble, which may mask

compensating biases and different dynamical responses across models and may be sensitive to model selection. We propose to a) expand the discussion of the maps/sections of individual ESMs already included in the Supplementary Information (SI) of the original manuscript, b) add individual ESMs when presenting OMZ volume and OMZ volume trends in the main text (new Figures GAC4, GAC5, GAC6), and c) compare the effects of water mass fraction changes (from OMP analysis) across individual models (Figure GAC3). Overall, we find that most models exhibit the same regional pattern of oxygen trends and regimes of OMZ volume trends as the MMM, with varying amplitudes of features across models. This will also be discussed in Discussion section 4.4.

We are confident that these revisions will address the concerns of the reviewers. Below, we provide specific details and figures for these revisions, and we address additional reviewer comments.

[Figure]

Figure GAC (General Author Comment) 1: Multi-model mean (MMM) stratification trends under SSP5-8.5 scenario forcing (1915-2100). Stratification trends (a) between 100 and 200m, and (b) at 65E. (c) dissolved oxygen trends at 65E. (b,c) Solid black contours represent 20, 60 and 150 µmol/kg oxygen. Dashed black contours highlight salinity signature of Subtropical Underwater (STUW). Results are stippled where less than 75% (6/8) of models agree on sign of trend.

[Figure]

Figure GAC2: Multi-model mean (MMM) export of organic carbon at 100 m trends under SSP5-8.5 scenario forcing (1915-2100). Results are stippled where less than 75% (6/7) of models agree on sign of trend.

[Figure]

Figure GAC3: Changes in dissolved oxygen accounted for by mixing model analysis. Total oxygen changes (simulated by ESMs) versus oxygen changes attributed to shifts in water mass composition in the OMP mixing mode for the (a) western South Equatorial Current (SEC) and (b) northern Arabian Sea. Results in the western SEC averaged between 40-80E, 5-15S. Results in the northern Arabian Sea averaged between 62-70E, 20-26N. Solid markers are the mean of 50 runs (light markers) of the OMP analysis with perturbed source water properties (See end of general response for details on new methods). Black dashed line marks 1-to-1 relationship, gray dashed lines mark 5 umol/kg deviations from 1-to-1.

**New method to be added to Optimum Multiparameter (OMP, section 2.2.4) analysis for sensitivity analysis and used to produce Figure GAC3 above:**

"From the mixing model results, we quantify the change in oxygen accounted for by shifts in water mass composition for key regions: $\Delta O_{mixing} = \sum_i (f_{i,1} O_{i,1} - f_{i,0} O_{i,0})$, where $f_i$ and $O_i$ are the fraction and oxygen concentration of source water type $i$, for future (1) and historical (0) states. Oxygen is allowed to be remineralized, so $\Delta O_{mixing}$ need not approximate the total simulated change in oxygen. To test the sensitivity of the mixing model results to small changes in the definition of source water types and potential density layer, we average 50 realizations of the experiment applying random perturbations to source water locations and density values. Source water locations are perturbed by up to 5 degrees in latitude and longitude for the South Equatorial Current region and 2 degrees for the Arabian Sea. For both locations, the value of the potential density layer is perturbed by up to 0.1."

We note regarding the OMP analysis that we lowered the weight on oxygen in the optimization compared to the submitted manuscript (lowered from 20% the weight of temperature and salinity to 2%) to more appropriately account for the range of oxygen values in the analysis compared to the range and temperature and salinities (see Appendix A of submitted manuscript for technical details of OMP).

[Figure]

Figure GAC4: (**To replace Figure 2 in main text**) OMZ volume taken between 0-1000 m in the (a) Indian Ocean and sub-basins (b) Arabian Sea and (c) Bay of Bengal for the multi-model mean averaged over 1950-2015 (MMM; black) and observed climatology (WOA18; red). Shading represents one standard deviation of the model spread. Individual ESMs shown in colored curves. Southern boundary of Arabian Sea and Bay of Bengal are at 5N.

[Figure]

Figure GAC5: (**To replace Figure 3 in main text**) Multi-model mean thermocline OMZ volume changes (between 0 - 1000 m) under SSP5-8.5 scenario forcing (2015-2100). (a) $V_{o2}^{1000}$ trends as a function of oxygen threshold for the Indian Ocean. The 20, 60 and 150 umol/kg thresholds bounding OMZ20, OMZ60 and OMZ150 are indicated with grey dashed lines. (b) Time series of $V_{o2}^{1000}$ anomaly from 1950-2100 (anomaly referenced to 1950-2015 mean) for OMZ60 (dark blue) and OMZ150 (teal). Shading represents one standard deviation of model spread.

[Figure]

Figure GAC6: (**To be added to supplementary information**) Multi-model mean thermocline OMZ volume changes (between 0 - 1000 m) under SSP5-8.5 scenario forcing (2015-2100). $V_{o2}^{1000}$ trends as a function of oxygen threshold for the (a) Arabian Sea and (b) Bay of Bengal with OMZ20 volumes indicated for each basin. (c) Time series of $V_{o2}^{1000}$ anomaly from 1950-2100 for OMZ20 volumes in the Arabian Sea and Bay of Bengal. Shading represents one standard deviation of model spread.

**Reviewer 3:**

I very much enjoyed reading this comprehensively written and well structured manuscript by Sam Ditkovsky et al. What is presented is an analysis of the general ventilation pathways in the Indian Ocean and which are "tailored" to the specifics of oxygen, including OMZ regions, putting an emphasis on oxygen concentration thresholds. On the whole, I agree with the approaches, and I like the analysis presented. However, some things are unclear to me and I would like the authors to comment on them. The analysis focusses on section/surfaces – what has been ignored in the study, but I assume is highly relevant when it comes to understanding the drivers of oxygen variability, is the physics of ventilation in particular in the southern source regions. In these regions a coexistence of stratified Central Waters (Ekman pumping driven subduction) with lateral flux dominated Mode waters exists (e.g. apparent in the global assessment of Hanawa & Talley, 2001, or specifically addressed in the Karstensen & Tomczak 1998 paper or). Given the specific role of Mode Waters in thermocline ventilation (in the Indian Ocean formed only in the southern hemisphere). I assume not addressing Mode Water and not contrasting them to Central Water may prompt wrong conclusions on the drivers/sources of variability – also for the Indian Ocean as a whole (because these water masses are a major source) . Can you show that this distinction between Mode and Central water does not matter?

Thank you for the thoughtful comment. As shown in Karstensen & Tomczak (1998), despite having distinct formation mechanisms, Indian Central Waters and Subantarctic Mode Waters have similar temperature and salinity signatures, and thus occupy the same density layer. That study finds that pure Central Waters only exist at midlatitudes, suggesting that by the time these waters reach the subtropical Indian Ocean thermocline (covered in our study), they are mixed together, and together they form the thermocline oxygen maximum layer. Thus we treat them as a mixed water mass of 'Central and Mode Waters' in the context of our study. Since we have no reliable way of distinguishing changes in Central Waters versus Mode Waters, we must remain agnostic as to whether the weakened thermocline ventilation in the CMIP6 experiments derives from changes in Ekman pumping or changes in Mode Water formation. We will clarify this in section 4.4 where we discuss the slowdown of southern source waters (line 434).

Within this context I wonder if analyzing the "southern source" by integrating over a vertical section at 30S only is maybe too simple? – for example the winter outcrop (determines area of permanent subduction) is not strictly zonal and thus the full ventilation signal is not accounted for by making use of a zonal section. This may get worse in ESM's that, with climate trends may show shifts in outcrop density change/trend over time? Have you considered that?

Indeed, we need to be careful in interpreting transport changes across 30S. A zonal section at 30S should comprehensively capture the northward transport of Central Waters and Subantarctic Mode Waters, as well as Antarctic Intermediate Waters, into the Indian Ocean. However, this section will miss the majority of Subtropical Underwater, which is mostly subducted north of 30S. We propose to add a comment on this in methods section 2.2.3. Additionally, any discussion of the transport across 30S in the revised manuscript will reflect its incompleteness of capturing the southern source waters.

A possible consequence of this is that the transport timeseries presented in Figure 7 (submitted manuscript) is incomplete. In particular, the ratio of Southern Pathway to ITF transport should likely increase more than is represented. This may be connected to a point brought up by Reviewer 1 and addressed in Figure R1AC3, which shows that the oxygen supply from the Southern Pathway and ITF collectively does not increase, given how we represent the pathways. To avoid confusion here, we remove Figure 7b from the text.

Also, I wonder if the AOU/ideal age comparison isn't a bit too simple to deduce biology from it? (Line 150 etc.).
This is because AOU is a property that reflects in my view three processes:
1) respiration (biology, depth and eventually region depending)
2) mixing of water properties in the interior
3) mixing of the imprint of the saturation value in the respective outcrop region of the various water masses
To explain what I mean - let's ignore 1) and just look at combinations of 2) & 3). For simplicity assume two water masses are 3 years old and have similar TS in their formation region – a mixing of the two would be invisible and the ideal age equals the oxygen propagation time (3 years). For this case all assumption are OK. However, if two water masses still are 3 years old but start with different TS and thus concentrations (saturation) in their formation region a mixing signal is seen in AOU but no signal in "ideal age" (this will be still 3 years). This change in AOU now is interpreted as a residual and thus "biology" (1)) despite the fact that in this thinking experiment no biology (1)) was considered. Given the complexity of water masses in the Indian Ocean and the wide range of oxygen concentrations and therefore oxygen gradients and therefore oxygen fluxes, the above process I assume may be significant and messes up the correlation and deduction of biology and driver of variability?

This is a good point, and the original language in lines 155-156 was misleading. In our comparison of AOU and ideal age (Figures 5,6), we only interpret an agreement between AOU and ideal age changes as a suggestion of ventilation changes (rather than interpreting any disagreement as biology changes). To avoid confusion, we plan to

remove the Pearson Correlation methods and Figure 6 panels d,e,f from the revised manuscript. See our General Response #1 for details on proposed additions to the manuscript which will address the role of changes in the biological sink.

Line 172: note that for an "Optimum" Multiparameter Analysis, at least one parameter more than the number of source water types is needed. This is because the term "Optimum" refers to the "non-negative" constrain which takes away on degree of freedom (see Lawson and Henson algorithm mentioned in Tomczak & Large). Is that a problem for your case? (>3 source water types?

There are perhaps two components here. The first is the allowed number of source water types relative to constraints/parameters. In the application of water masses, conservation of mass adds an additional constraint (eg Tomczak 1981, Prog. Ocean.), so one can actually have up to *n* source water types for *n* non-mass parameters. In our study, we use as parameters potential temperature, salinity, oxygen, and AOU (and mass). One can argue whether or not AOU should count as an independent parameter here, but we will take the conservative stance of claiming to use only 3 parameters (T, S, O2) + mass, allowing for 3 source water types.

The second component here is whether 3 source water types is sufficient for describing the Indian Ocean. We found that each of our evaluation regions (the tropical Indian Ocean and the Arabian Sea) could be well approximated by just 3 source water types. This relies somewhat on the assumption that the system can be explained to first order only by isopycnal transport/mixing. For example, rather than capturing the full linear relationship of the Southern Pathway source waters by using both Antarctic Intermediate Water and Subtropical Underwater, we use a single southern source sampled at the same potential density on which we evaluate the water mass composition of the tropical Indian Ocean.

Line 224: is it known why ESMs simulate higher oxygen levels? Could it be that shallow subduction occurs in the Arabian Sea in the models? (the Ekman pumping during monsoon would support that).
A study by Schmidt et al. (2021, Ocean Science) investigated the Arabian Sea oxygen bias for CMIP5 models (and CMIP6 models likely experience similar issues). They attribute the model biases primarily to excessive ventilation from southern source waters and the Red Sea and Persian Gulf outflows. We find that this explanation is consistent with what we see in the CMIP6 ESMs.

Figure 5: you indicate the expansion of water masses by dashed line. How can I envision this expansion? Say AAIW is defined by the salinity minimum – what does expansion mean? A density range? The 95% contour of AAIW content?

The dashed contours in Figure 5 simply outline the historical locations of AAIW and STUW cores, defined using approximate salinity minimum and maximum values, respectively. We do not suggest in this figure how these salinity features evolve over time, only the evolution of oxygen within and around these salinity features. We will clarify in the caption that these contours represent historical locations of the water masses, not an evolution.

Line 327-329 or Line 353-355: you report about the increase/decrease of water masses overtime. Operating with changing source water mass characteristics is a challenge in water mass mixing analysis – simply because, from a mixing model point of view, each changing source water is introducing an additional water mass to the ocean interior while the water with the former characteristics still exist and contribute to the mixing. How do you deal with that? (maybe I overlooked it but do you list the source waters somewhere?)

We agree that this is a fundamental challenge of the analysis, with perhaps no perfect solution. However, we try to mitigate this effect here by solving the mixing model for climatological fields, rather than evolving fields. In this case, we use 1950-2015 as a historical average. We then compute a climatological year 2100 by perturbing the historical mean using linear trends over the SSP5 experiment.

In the revised manuscript, we propose to present the composite of many runs of the OMP analysis with small perturbations to source water type properties (detailed in the Appendix to the General Response). The results show little sensitivity to these perturbations. While this does not capture the evolution of source water properties in time, it does provide an estimate of the sensitivity of the analysis to small changes in source water properties.

Line 420: An additional fact on the Atlantic OMZ ventilation is the significant source of South Atlantic source waters on ventilating the North Atlantic OMZ (e.g. evident in the TS properties but also from subduction estimates). You may want to also consider this in your "first glance on Atlantic" discussions.

Thank you for the suggestion. Some previous modeling studies suggest that subduction rates of Mode and Intermediate Waters from the South Atlantic hold steady in ESM projections, but that they ventilate at lighter densities (eg Goes et al., 2008, Downes et al., 2009). Given that the Atlantic OMZ is relatively shallow, and that competing pathways (ie deep waters) are likely to slow, this seems to be a plausible hypothesis. We propose to add this to our discussion in section 4.3.

---

## Author Response (AR2)

We thank the reviewers and associate editor for their efforts on behalf of our manuscript. We submit a final version which addresses the following technical correction from Reviewer 3:

"Please note that the use of Optimum MP (Tomczak and Large 1989) versus MP (Tomczak 1981) is incorrect and requires correction. "Optimum" indicates that the results of the analysis are "non-negative" - with the expense in losing one degree of freedom (4 parameters can resolve up to 4 source waters). Technically correct would be that with 4 parameters only 3 source waters can be resolved to ensure the system is "overdetermined" (not only determined).
Furthermore, if one uses the "extended OMP" one more degree of freedom is lost because a new unknown (amount of remineralized material) is added.
Please adjust the references and methods naming accordingly."

We agree with the distinction made by the reviewer and have changed the use of 'OMP' to a more generic 'mixture model' where relevant. We explain the differences between our implementation and the traditional usage of OMP in the appendix which details the method:

"We note that the mixing model used in this study differs in several ways from the traditional extended OMP. We solve a 'mixing-triangle' of three source water types, rather than four as used in, for example, Tomczak Jr (1981). Adding an extra degree of freedom to track remineralization, we thus have a total of four degrees of freedom. Typically, remineralization is tracked as the conversion of oxygen to phosphate and nitrate using the Redfield Ratio. Since we do not have access to full nutrient fields for the ESM experiments used in this study, we simply allow for the conversion of oxygen to AOU with a ratio of $-1$. As constraints, we use potential temperature, salinity, dissolved oxygen, AOU and mass conservation. Since AOU is a function of dissolved oxygen, potential temperature and salinity, we have four unique constraints. Thus, our mixture model is equivalent to a determined set of linear equations, rather than an over-determined set (i.e. 'Multiparameter Analysis', rather than 'Optimum Multiparameter Analysis'; Tomczak Jr, 1981; Tomczak and Large, 1989). However, since AOU must also be included in the calculation, we leverage the least squares method used in solving the over-determined problem."